# The Spc105/Kre28 complex promotes mitotic error correction by outer kinetochore recruitment of Ipl1/Sli15

Alexander Dudziak[1], Richard Pleuger [ID][1], Jasmin Schmidt[1], Frederik Hamm[1], Sharvari Tendulkar[1], Karolin Jänen[1], Ingrid R Vetter [ID][2], Sylvia Singh[3], Josef Fischböck[3], Franz Herzog[3,4] & Stefan Westermann [ID][1✉]

## Abstract

**Kinetochores link chromosomes to dynamic microtubules of the mitotic spindle. To ensure equal chromosome segregation, sister chromatids must achieve biorientation. The conserved kinase Aurora B phosphorylates outer kinetochore proteins on attachments lacking tension, allowing the re-establishment of new connections until biorientation is achieved. Aurora B localizes to the centromere as part of the chromosomal passenger complex (CPC), but the underlying recruitment pathways can be eliminated without disrupting biorientation. It therefore remains unclear how the kinase operates during error correction. Here, we identify the conserved Spc105/Kre28 complex as an outer kinetochore receptor of the Aurora kinase Ipl1 and its activator Sli15 in *Saccharomyces cerevisiae*. We show that mutations in the helix bundle domain of Spc105/Kre28 impair mitotic error correction, resembling the effects of *ipl1* or *sli15* mutants. The defects can be suppressed by the artificial recruitment of Ipl1. In biochemical experiments, Ipl1/Sli15 directly associates with Spc105/Kre28, and a conserved segment in the Sli15 central domain is crucially involved in the binding mechanism. These results have important implications for the mechanism of tension-dependent error correction during chromosome biorientation.**

**Keywords** Chromosome Biorientation; Error Correction; Chromosomal Passenger Complex; KMN Network; Signaling
**Subject Categories** Cell Adhesion, Polarity & Cytoskeleton; Cell Cycle

## Introduction

Faithful chromosome segregation during mitosis depends on the physical interaction between highly dynamic microtubules of the mitotic spindle and kinetochores which assemble on centromeric DNA of each sister chromatid. Kinetochores are megadalton-sized proteinaceous structures consisting of multiple subcomplexes that fulfill different functions and assemble in a hierarchical manner. The constitutive centromere-associated network (CCAN) is the main constituent of the inner kinetochore (Dendooven et al, 2023; Pesenti et al, 2022; Yatskevich et al, 2022) which is recruited to centromeres through the interaction of the Mif2[CENP-C] subunit with centromeric nucleosomes (Kato et al, 2013; Xiao et al, 2017). The CCAN consecutively recruits subcomplexes of the outer kinetochore generally termed as KMN network which consists of the Spc105[Knl1]/Kre28[Zwint] complex, Mtw1 complex (Mtw1, Dsn1, Nnf1, Nsl1), and Ndc80 complex (Ndc80, Nuf2, Spc24, Spc25) (Polley et al, 2024; Yatskevich et al, 2024). In this context, the Mtw1 complex serves as a scaffold which connects the other KMN subcomplexes with the inner kinetochore by direct association with Mif2[CENP-C] and Ame1[CENP-U] (Dimitrova et al, 2016; Hornung et al, 2014; Musacchio and Desai, 2017; Petrovic et al, 2016; Weir et al, 2016). The highly elongated Ndc80 complex is the main microtubule receptor of the kinetochore which binds to the mitotic spindle through the calponin homology (CH) domains at the N-termini of the Ndc80 and Nuf2 subunits (Ciferri et al, 2008; Wei et al, 2007; Wilson-Kubalek et al, 2008). The Ndc80/Nuf2 head also recruits the checkpoint kinase Mps1 to the kinetochore (Parnell et al, 2024; Pleuger et al, 2024; Zahm & Harrison, 2024). The RWD domains of the Spc24 and Spc25 subunits connect the Ndc80 complex with the Mtw1 complex (Dimitrova et al, 2016; Malvezzi et al, 2013; Petrovic et al, 2016). Furthermore, the Mtw1 complex recruits Spc105[Knl1]/Kre28[Zwint] to the kinetochore, even though the molecular mechanisms differ between mammals and yeast (Ghodgaonkar-Steger et al, 2020; Petrovic et al, 2014; Roy et al, 2022). It has been well-established that Spc105[Knl1] plays a crucial role as a signaling platform for the spindle assembly checkpoint (SAC) after phosphorylation of MELT repeats by the conserved kinase Mps1 (London et al, 2012; Shepperd et al, 2012; Yamagishi et al, 2012). Furthermore, Spc105[Knl1] is required for the recruitment of different protein phosphatases which are involved in SAC

[1]Department of Molecular Genetics I, Faculty of Biology, Center of Medical Biotechnology, University of Duisburg-Essen, Universitätsstrasse 5, 45117 Essen, Germany. [2]Department of Mechanistic Cell Biology, Max-Planck-Institute of Molecular Physiology, Otto-Hahn-Straße 11, 44227 Dortmund, Germany. [3]Gene Center Munich and Department of Biochemistry, Ludwig-Maximilians-Universität München, Feodor-Lynen-Str. 25, 81377 Munich, Germany. [4]Institute Krems Bioanalytics, IMC University of Applied Sciences, Krems, Piaristengasse 1, A-3500 Krems, Austria. ✉E-mail: Stefan.Westermann@uni-due.de

silencing and modulation of kinetochore–microtubule attachments (Meadows et al, 2011; Nijenhuis et al, 2014; Rosenberg et al, 2011). In contrast, relatively little is known about Kre28[Zwint] (also referred to as YDR532C according to its systematic name). Kre28 was identified as budding yeast homolog of the metazoan protein Zwint (Zeste White interacting protein) due its sequence similarity and, similar to Zwint, Kre28 forms a complex with Spc105[Knl1] (Pagliuca et al, 2009). In contrast to mammalian cells, where the tandem RWD domains of Knl1 directly associate with the Mtw1 complex (Petrovic et al, 2014), S. cerevisiae Kre28 itself binds to the Mtw1 complex and by this recruits Spc105[Knl1] to the kinetochore (Ghodgaonkar-Steger et al, 2020; Roy et al, 2022). Both, the interaction with Spc105[Knl1] and with the Mtw1 complex is mediated by coiled-coil elements in the central region of Kre28 (Ghodgaonkar-Steger et al, 2020; Roy et al, 2022). Furthermore, the C-terminus of Kre28 (residues 299–386) forms an RWD domain whose function is unknown so far (Tromer et al, 2019). Kre28 is essential for viability of yeast cells (Pagliuca et al, 2009). However, it is unclear whether this is just a consequence of loss of Spc105[Knl1] from kinetochores or whether Kre28[Zwint] has additional essential functions. For instance, it was proposed that Spc105[Knl1]/Kre28[Zwint] might be required for kinetochore localization of different microtubule-associated proteins (Pagliuca et al, 2009).

The conserved protein kinase Ipl1[Aurora B] is an essential part of the error correction machinery which phosphorylates outer kinetochore substrates in order to release kinetochore–microtubule attachments that lack tension (Tanaka and Zhang, 2022). Ipl1[Aurora B] acts in a complex with Sli15[INCENP], Bir1[Survivin] and Nbl1[Borealin], together termed as chromosomal passenger complex (CPC). The N-terminal CEN box of Sli15[INCENP] associates with Bir1[Survivin] and Nbl1[Borealin] (Jeyaprakash et al, 2007), whereas its C-terminal IN box binds to the kinase domain of Ipl1[Aurora B] (Sessa et al, 2005). The CPC is targeted to the inner and kinetochore-proximal centromere by different mechanisms which depend on the haspin kinase, the Bub1 kinase and Sgo1/Sgo2 (Hindriksen et al, 2017). However, different studies have demonstrated that inner and kinetochore-proximal centromere localization of Ipl1[Aurora B] is dispensable for the establishment of biorientation. For instance, an N-terminal deletion of Sli15, preventing inner centromere recruitment of the CPC through Bir1, still supports the formation of bioriented attachments (Campbell and Desai, 2013), suggesting that additional mechanisms can target the CPC to the kinetochore. Indeed, two recent studies identified the inner kinetochore COMA complex as a Bir1-independent Ipl1/Sli15 receptor (Fischbock-Halwachs et al, 2019; García-Rodríguez et al, 2019). Furthermore, studies in human cells have also suggested the existence of an additional pool of Aurora B at the kinetochore, which might be relevant for the phosphorylation of outer kinetochore proteins such as components of the KMN network (Broad et al, 2020; Hadders et al, 2020).

In this study, we provide a detailed molecular analysis of the KMN subunit Kre28[Zwint]. We identified a conserved Zwint helix in Kre28 whose lack leads to fatal chromosome segregation defects when cells are forced to reform their spindle apparatus after nocodazole washout. Artificial recruitment of Ipl1 to the outer kinetochore is sufficient to suppress this phenotype. Using protein biochemistry, we demonstrate that Spc105[Knl1]/Kre28[Zwint] can directly recruit Ipl1[Aurora B]/Sli15[INCENP] in vitro and that a conserved segment of the Sli15 central domain is involved in the binding mechanism. Thus, we have identified Spc105/Kre28 as the elusive outer kinetochore receptor of Ipl1.

# Results

## Characterization of novel mutant alleles in the outer kinetochore subunit Kre28[Zwint]

To gain insight into the organization of Spc105/Kre28 we performed AF2 structure predictions using the C-terminal part of Spc105 (omitting the disordered residues 1–489) and full-length Kre28 (Fig. 1A). The model shows that the Spc105/Kre28 heterodimer is formed via two interfaces: At the C-terminus of Kre28 a single RWD domain (residues 299–385) is paired with the N-RWD domain of Spc105. Together with short alpha-helical segments just preceding these RWD domains, this part resembles heterodimeric RWD domain arrangements found in other kinetochore subunits like Spc24/Spc25 (Schmitzberger and Harrison, 2012). The N-terminal parts, on the other hand, engage into a helical bundle (Fig. 1A). This is in agreement with previous studies that have established that Kre28 associates with Spc105 mainly via coiled-coil segments formed by residues 127–201 (Roy et al, 2022), while another segment (229–259) is required for binding to the Mtw1 complex (Ghodgaonkar-Steger et al, 2020). Deletion of these segments is incompatible with KMN assembly and yields lethal phenotypes in yeast. Interestingly, multiple sequence alignments show that an alpha-helical segment preceding the first Kre28 coiled-coil domain (residues 77–113, hereafter referred to as the Zwint helix) is among the most conserved segments of the protein (Fig. 1B). By contrast, the C-terminus is not well conserved and human Zwint, for example, lacks an RWD domain (Tromer et al, 2019).

To analyze the contribution of Zwint helix and RWD domain to Kre28 function in cells, we replaced the endogenous KRE28 gene with Flag-tagged wild-type or mutant alleles in diploid cells (Fig. 1C). Tetrad dissection showed that neither kre28[ΔZwint] nor kre28[ΔRWD] alone impaired viability. However, haploid spores of the kre28[ΔZwint ΔRWD] double mutant were never recovered, demonstrating that simultaneous elimination of these segments is lethal (Fig. 1C). Western blotting confirmed that wild-type and mutant proteins were expressed at comparable levels (Fig. 1D). Interestingly, serial dilution assays of the haploid strains revealed that kre28[ΔZwint] mutants fully supported growth at 30 °C, but were inviable in the presence of the microtubule-depolymerizing drug benomyl and additionally displayed cold sensitivity (Fig. 1E). By contrast, the deletion of the C-terminal RWD domain of Kre28 did not cause a growth phenotype. We found that not only the complete removal of the Zwint helix but also selected smaller deletions or point mutations of conserved residues caused benomyl hypersensitivity. The most pronounced phenotypes were obtained for truncations eliminating the c-terminal part of the helix spanning residues 103–113, as well as for point mutations in this region (Fig. EV1A–D).

## The kre28[ΔZwint] mutant is defective in mitotic error correction of kinetochore–microtubule attachments

The pronounced benomyl hypersensitivity of the kre28[ΔZwint] mutant may be caused by a defect in the spindle assembly checkpoint (SAC) or a different problem with chromosome segregation. To distinguish between these possibilities, we analyzed Pds1[Securin] levels in wild-type and kre28[ΔZwint] strains after cells were released from a G1 arrest into medium containing the microtubule-depolymerizing drug nocodazole. Both, wild-type and kre28[ΔZwint] mutant cells stabilized Pds1 levels, while a mad1Δ control strain failed to do so (Fig. 2A). Thus, kre28[ΔZwint] cells have a functional SAC response to a lack of kinetochore–microtubule

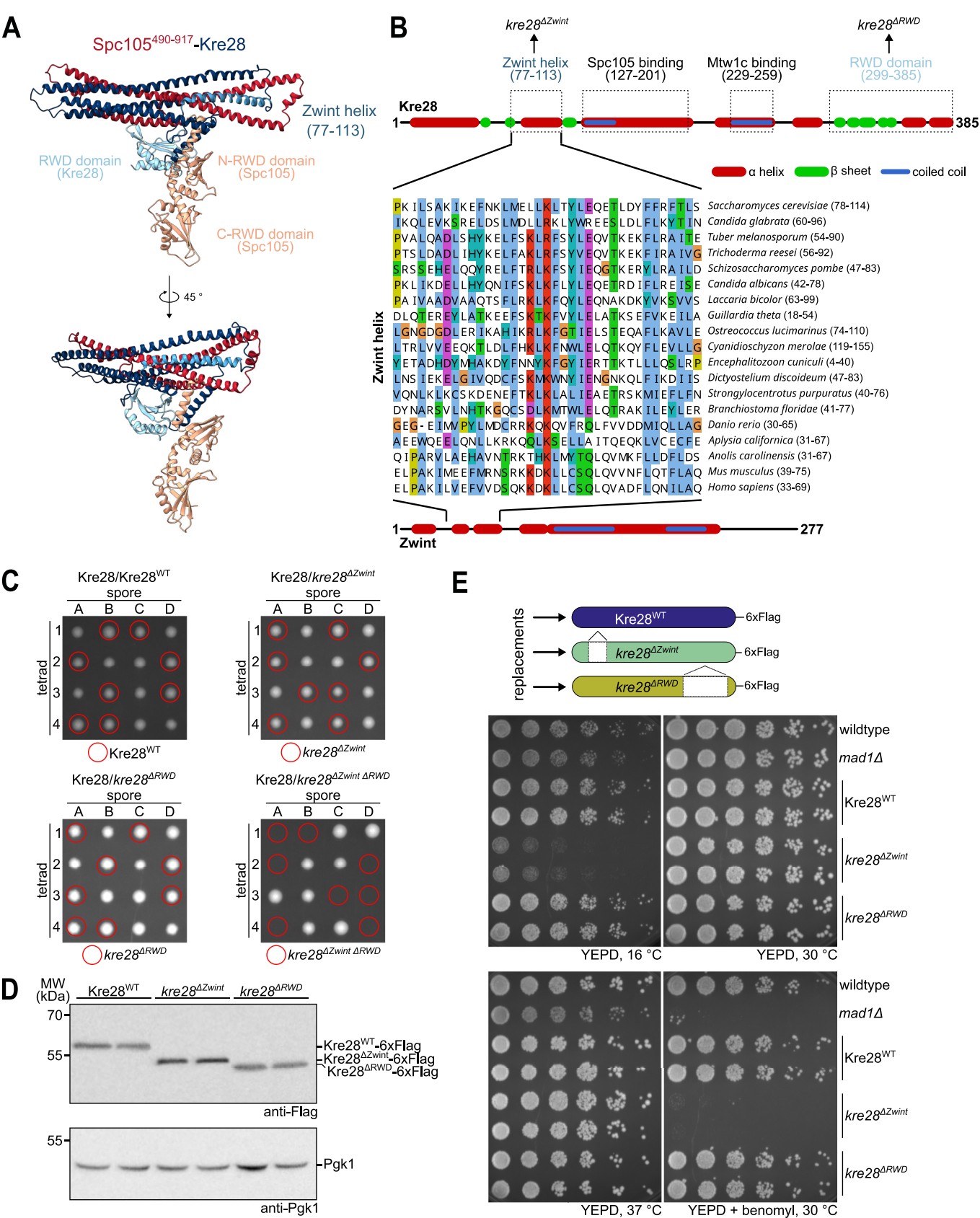

**Figure 1. Mutation of the Kre28 helical bundle renders yeast cells inviable in the presence of benomyl.**

(A) AlphaFold2 structure prediction of the Spc105[490-917]/Kre28 heterodimer. The unstructured N-terminal half of Spc105 was omitted from the structure prediction. Functionally relevant regions of the proteins such as Zwint or RWD domains are labeled in distinct colors. (B) Secondary structure prediction of *S. cerevisiae* Zwint (top), *H. sapiens* Zwint (bottom), and multiple sequence alignment of the so-called Zwint helix of related eukaryotic proteins (middle). Functional regions of Kre28 required for Spc105 (Roy et al, 2022), Mtw1c binding (Ghodgaonkar-Steger et al, 2020), the C-terminal RWD domain (Tromer et al, 2019), and the newly identified Zwint helix are marked. (C) Representative images of dissection plates of diploid heterozygous Kre28/Kre28-6xFlag yeast strains with different 6xFlag-tagged alleles (WT, ΔZwint, ΔRWD, or ΔZwint ΔRWD). Strains were sporulated and dissected, and the plates were incubated at 30 °C for 3 days. Spores were subsequently analyzed regarding their genotype. While haploid strains with single deletion of either the Zwint helix or the RWD domain are viable, simultaneous deletion of both domains does not support the viability of haploid yeasts. (D) Western blot analysis to compare protein expression levels of Kre28^WT, kre28^ΔZwint, and kre28^ΔRWD in haploid yeast strains. Pgk1 was used as a loading control. (E) Serial dilution assay of haploid yeast strains with different Kre28 alleles. Strains with the indicated genotypes were serially diluted, spotted on YEPD or YEPD + benomyl plates, and incubated at the indicated temperatures. A *mad1Δ* strain was used as a control for benomyl hypersensitivity. Source data are available online for this figure.

attachments. To define the genetic relationship between the *kre28^ΔZwint* mutant and the mitotic checkpoint more closely, we deleted the *MAD1* gene in the *KRE28* wild-type or *kre28^ΔZwint* mutant background. Abolishing the SAC aggravated the growth defect of the *kre28^ΔZwint* mutant, an effect that was particularly apparent at temperatures of 25 °C and 16 °C (Fig. 2B). Thus, the benomyl hypersensitivity of the *kre28^ΔZwint* mutant is not caused by a general mitotic checkpoint defect, and indeed, the SAC is required to support the viability of this mutant.

While *kre28^ΔZwint* mutant cells arrested efficiently in response to nocodazole, we found that they nevertheless very quickly lost viability when cells were plated from nocodazole-containing medium onto YEPD plates (Appendix Fig. S1). To determine the reason for the rapid loss in viability, we arrested wild-type and mutant cells with nocodazole, then washed out the drug to allow spindle reformation and chromosome attachment and followed the progression through the first cell division. Analysis of Pds1 degradation kinetics showed that *kre28^ΔZwint* mutant cells initiated anaphase—indicated by the decline in Pds1 levels—with the same timing as wild-type cells (Fig. 2C). The lack of a delay indicated that the SAC was not activated under these conditions. In the same experimental setup, we analyzed the distribution of fluorescently labeled chromosome XII (CEN12-GFP) following the nocodazole washout (Fig. 2D). Strikingly, *kre28^ΔZwint* mutant cells missegregated chromosome XII in ~70% of all anaphases, while wild-type cells displayed an error rate of less than 5% (Fig. 2E). This high rate of missegregation can explain the pronounced benomyl hypersensitivity and the rapid loss of viability after exposure to nocodazole.

The combination of phenotypes displayed by the *kre28^ΔZwint* mutant —a functional SAC in response to microtubule depolymerization, but pronounced chromosome missegregation without checkpoint activation following spindle reformation—is reminiscent of mutants in the tension-dependent error correction pathway, such as *ipl1*, *sli15*, or *sgo1* (Biggins and Murray, 2001; Indjeian et al, 2005). In genetic crosses, we found that the *kre28^ΔZwint* mutant was highly sensitive to further perturbations of the error correction machinery: Its combination with the temperature-sensitive *ipl1-1* allele resulted in extremely poorly growing spores (Fig. 2F) and its combination with *sli15-3* or *sgo1Δ* was synthetic lethal. The genetic interactions of the *kre28^ΔZwint* allele are summarized in Fig. 2G. Deletion of the Kre28 Zwint helix is synthetically lethal or aggravates growth defects when the function of the microtubule-binding Ndc80 complex (*ndc80-1*) and Dam1 complex (*dam1-1*, *dam1-9*, and *duo1^ΔSxIP*) is compromised. Furthermore, either direct perturbation of the CPC (*ipl1-1* and *sli15-3*) or affecting its centromere (*sgo1Δ*) or inner kinetochore recruitment pathways (*ctf19^ΔC-RWD*) dramatically reduced cell growth in combination with the *kre28* mutant allele.

## The *kre28^ΔZwint* mutant reduces the level of Spc105/Kre28 at kinetochore clusters in vivo

To further define the molecular basis of the *kre28^ΔZwint* phenotype, we analyzed outer kinetochore composition in wild-type and mutant cells by live-cell imaging. The fluorescence intensity of the KMN components Nuf2-GFP (Ndc80 complex) and Mtw1-GFP (Mtw1 complex) at metaphase and anaphase kinetochore clusters was unaffected by the *kre28^ΔZwint* mutation (Fig. 3A). By contrast, we noticed a reduction of Spc105-GFP intensity at metaphase kinetochores to ~70% of the wild-type level and found that the apparent increase in cluster intensity between metaphase and anaphase was reduced relative to the wild-type (Fig. 3B). These differences were not caused by changes in protein levels, as judged by western blotting (Fig. EV2A,B). Since we found that a C-terminal GFP fusion caused a reduced protein level of the *kre28^ΔZwint* mutant (Fig. EV2C), we constructed and analyzed N-terminal GFP-fusions of wild-type and mutant alleles. GFP-Kre28^ΔZwint was present at near wild-type levels in western blots (Fig. EV2B). Similar to Spc105-GFP, the GFP-*kre28^ΔZwint* mutant displayed a moderate reduction in metaphase kinetochore cluster intensity and a less pronounced increase in apparent anaphase cluster intensity (Fig. 3B). We conclude that the *kre28^ΔZwint* mutation has a specific effect on the KMN network in vivo: It reduces the localization of Spc105/Kre28 to kinetochores without affecting the recruitment of Ndc80 and Mtw1 complexes.

Since Spc105 acts as a scaffold for the Mps1-dependent recruitment of Bub1/Bub3, we asked if the Spc105 reduction would have any consequences on downstream signaling. We found that Sgo1-GFP localization to kinetochores in prometaphase (which is dependent on Bub1) was not affected by *kre28^ΔZwint*, suggesting that the modest reduction in Spc105 does not have a pronounced effect on downstream signaling (Fig. EV2E). This is also consistent with the observation that this mutant has a functional SAC.

## Artificial recruitment of Ipl1 to Mtw1 suppresses the *kre28^ΔZwint* phenotype in vivo

If the *kre28^ΔZwint* mutant reduces the association of Spc105/Kre28 to kinetochores in vivo and the resulting phenotype resembles a lack of Ipl1 activity, then artificially increasing Ipl1 recruitment in the mutant may rescue the defect (Fig. 4A). We tested this hypothesis by constructing fusion proteins between the Mtw1 subunit (whose kinetochore level is unaffected by *kre28^ΔZwint*, see Fig. 3) and different C-terminal segments of the Sli15 protein. The fusions were either prepared with wild-type Sli15 sequences or with two

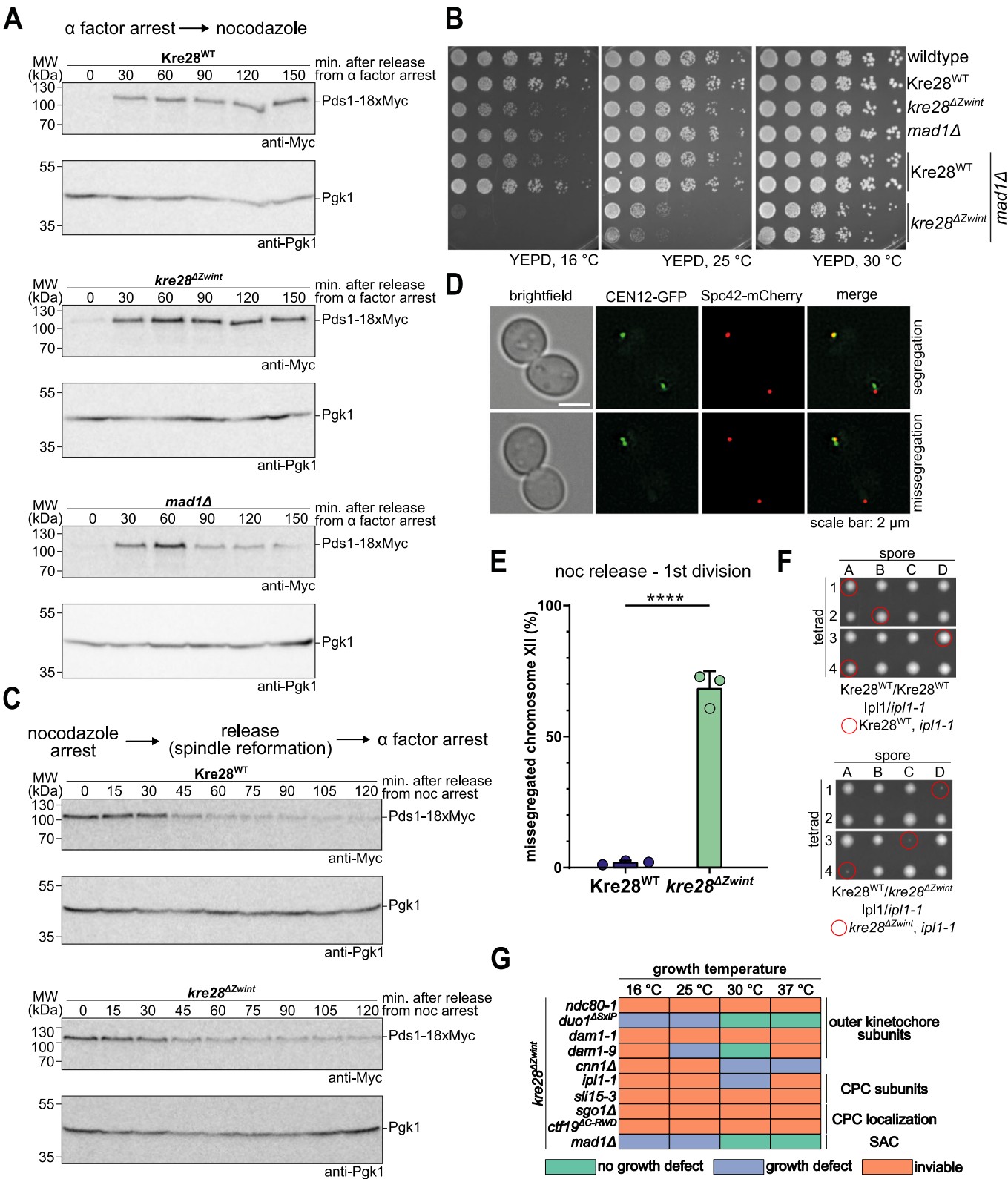

**A** α factor arrest ⟶ nocodazole

**B** wildtype / Kre28^WT / kre28^ΔZwint / mad1Δ / Kre28^WT / kre28^ΔZwint (mad1Δ)

YEPD, 16 °C    YEPD, 25 °C    YEPD, 30 °C

**D** brightfield    CEN12-GFP    Spc42-mCherry    merge

segregation / missegregation    scale bar: 2 µm

**E** noc release - 1st division

**F** spore

**C** nocodazole arrest ⟶ release (spindle reformation) ⟶ α factor arrest

**G** growth temperature

point mutations exchanging conserved amino acids in the IN box (W646G, F680A), which prevent Ipl1 binding (Sessa et al, 2005). The genes encoding the fusion proteins were integrated as extra copies at the *lys2* locus and expressed constitutively under control of the *MTW1* promoter (Fig. 4B; Appendix Fig. S2).

Serial dilution assays showed that expression of neither fusion protein alone caused any growth defects in a Kre28^WT strain background. Interestingly, Mtw1-Sli15^565-698 improved the growth of the *kre28^ΔZwint* mutant in the presence of benomyl. Moreover, the longer Mtw1-Sli15^516-698 fusion, encompassing the IN box and the

Figure 2.  The Kre28 helical bundle mutant displays defects in tension-dependent error correction.

(A) Analysis of cell cycle progression. Haploid yeast strains with either Kre28$^{WT}$, kre28$^{\Delta Zwint}$ or mad1$\Delta$ were first arrested in G1 using α factor and then released into medium containing nocodazole to depolymerize microtubules and tested for functionality of the spindle assembly checkpoint. Samples for western blot analysis were collected every 30 min and were analyzed for Pds1 levels as readout for mitotic progression. A mad1$\Delta$ strain was used as a positive control for spindle assembly checkpoint deficiency. (B) Serial dilution assay of haploid yeast strains with Kre28$^{WT}$ or kre28$^{\Delta Zwint}$ in a MAD1 or mad1$\Delta$ strain background. Strains with the indicated genotypes were serially diluted, spotted on YEPD or YEPD + benomyl plates, and incubated at the indicated temperatures. (C) Analysis of cell cycle progression after release from a nocodazole arrest. Kre28$^{WT}$ or kre28$^{\Delta Zwint}$ cells were arrested in mitosis using nocodazole and released into medium with α factor to arrest cells in the next G1 phase. Samples were collected every 15 min and Pds1 levels as marker for mitotic progression were analyzed by western blot. Pgk1 served as loading control. (D, E) Analysis of segregation of fluorescently labeled chromosome XII by live-cell microscopy. Kre28$^{WT}$ and kre28$^{\Delta Zwint}$ cells with fluorescently labeled chromosome XII (CEN12-GFP) and spindle pole bodies (Spc42-mCherry) were released from a nocodazole arrest and segregation of chromosome XII during the next anaphase was followed. Representative images of segregation and missegregation are shown in (D), a quantification of the missegregation rate is shown in (E). Data are derived from three independent experiments analyzing n > 100 cells per condition. The dots represent the data of each individual experiment. Error bars indicate standard deviation. P value was calculated by an unpaired t test. ****P < 0.0001. Exact P value for (E): P < 0.0001. (F) Representative images of dissection plates of diploid heterozygous Kre28/Kre28-6xFlag Ipl1/ ipl1-1 strains. Strains were sporulated and dissected and the plates were incubated at 30 °C for 3 days. Afterward, the spores were analyzed according to their genotype. Haploid strains with both kre28$^{\Delta Zwint}$ and ipl1-1 are viable but display severe growth retardation. (G) Summary of genetic interactions between the kre28$^{\Delta Zwint}$ allele and mutant alleles of outer kinetochore subunits (ndc80-1, duo1$^{\Delta SxIP}$, dam1-1 and dam1-9), the Ndc80c recruiter Cnn1 (cnn1$\Delta$), CPC subunits (ipl1-1, sli15-3), CPC recruiters (sgo1$\Delta$, ctf19$^{\Delta C-RWD}$), or a spindle assembly checkpoint component (mad1$\Delta$). Source data are available online for this figure.

single alpha-helix (SAH) domain, was even more effective in suppressing the benomyl hypersensitivity of the kre28$^{\Delta Zwint}$ mutant. Importantly, fusion proteins with point mutations in the IN box had no beneficial effect on the growth of kre28$^{\Delta Zwint}$ strains in the presence of benomyl, demonstrating the importance of Ipl1 recruitment to the outer kinetochore (Fig. 4C).

We further analyzed the effect of the two Mtw1-Sli15 fusions on kre28$^{\Delta Zwint}$ cells by scoring segregation of fluorescently labeled chromosome XII in the first division after release from a nocodazole arrest. As described before (see Fig. 2E), about 70% of kre28$^{\Delta Zwint}$ cells missegregated chromosome XII. In contrast, the rate of missegregation was significantly reduced in kre28$^{\Delta Zwint}$ cells expressing the Mtw1-Sli15 fusion proteins (Fig. 4D). Similar to the effect in the serial dilution assay, expression of the larger construct comprising Sli15 residues 516–698 showed a stronger suppression of the kre28$^{\Delta Zwint}$ phenotype. Notably, constitutive recruitment of Ipl1 the outer kinetochore via the Mtw1 fusion had no negative effect on chromosome biorientation in a wild-type strain background.

These data indicate that insufficient Ipl1/Sli15 at the outer kinetochore is a main cause of the phenotypes observed in kre28$^{\Delta Zwint}$ strains.

## Ipl1/Sli15 directly binds the KMN network via the Spc105/Kre28 subcomplex

We next asked whether the Ipl1/Sli15 complex may directly associate with the KMN network and specifically with Spc105/ Kre28. To this end, we co-expressed the ten KMN subunits from two separate baculoviruses in Sf9 insect cells and purified the network via a Flag tag at the C-terminus of the Spc105 subunit. This purification strategy allowed the single-step isolation of a stable full-length KMN network that co-eluted stoichiometrically during analytical size exclusion chromatography (SEC) (Fig. 5A). We evaluated the organization of the KMN network by Pt/C rotary shadowing electron microscopy. In electron micrographs the Ndc80 complex (N) was readily identifiable as a 50–60 nm long rod and its distal end was decorated by one or two appendages–formed by the KM complexes–which seemed to adopt a variety of different conformations relative to each other (Fig. 5B).

We then incubated the recombinant KMN network immobilized on anti-Flag M2 beads with bacterially expressed Ipl1/6xHis-Sli15 (Fig. 5C). Western blotting indicated that our Ipl1/6xHis-Sli15

preparation contained two major forms of Sli15, both recognized by the anti-His antibody. The faster migrating form is therefore a C-terminal truncation of Sli15. We found that wild-type Kre28 and Kre28$^{\Delta Zwint}$ KMN networks, but not empty control beads, bound the Ipl1/Sli15 complex in the pull-down experiment, with the full-length Sli15 form being detected preferentially in the pulldown (Fig. 5C). Whereas Ipl1/Sli15 appeared to bind equally well to both wild-type and mutant KMN networks, the association of Mtw1c/ Ndc80c to Spc105/Kre28 was slightly reduced in the Kre28$^{\Delta Zwint}$ mutant. Adjusting the pull-down protocol to prolonged incubation times, led to a more complete dissociation of Mtw1c/Ndc80c from Spc105/Kre28, while Ipl1/Sli15 binding was largely preserved (Fig. EV3A). Kinase assays showed that Spc105, Ndc80, and Dsn1 were major KMN substrates of Ipl1, as anticipated from previous work (de Regt et al, 2022). Interestingly, Ipl1 autophosphorylation—a readout for its activity—appeared to be increased in the presence of the KMN network (Fig. EV3B,C).

We next compared the ability of Ipl1/Sli15 to bind to the different subcomplexes of the KMN network by immobilizing Spc105-Flag/Kre28 (K subcomplex) or Mtw1c/Ndc80c-Flag (MN subcomplexes) separately on anti-Flag M2 beads. Western blotting showed that Ipl1/Sli15 interacted exclusively with Spc105/Kre28, but not with Ndc80c/Mtw1c under these conditions (Fig. 5D).

Our initial characterization showed that the Ipl1/Sli15 complex was autophosphorylated following expression and purification from E. coli. We tested the effect of Ipl1 autophosphorylation on the binding to immobilized Spc105/Kre28 by comparing an untreated control with samples including either ATP (to promote phosphorylation) or Lambda phosphatase (to eliminate phosphorylation). We found that inclusion of ATP strongly reduced the association of Ipl1/Sli15 with Spc105/Kre28, while addition of Lambda phosphatase promoted binding (Fig. 5E). Judged by protein band intensity, near stoichiometric amounts of Ipl1 seemed to bind to immobilized Spc105/Kre28 in the untreated or Lambda phosphatase-treated sample. Thus, we conclude that Ipl1/Sli15 binds recombinant KMN in vitro via the Spc105/Kre28 complex and that this interaction can be counteracted by Ipl1 autophosphorylation.

Unlike the checkpoint kinase Mps1, the Ipl1/Sli15 kinase complex is typically not recovered as a stable component of kinetochore purifications from yeast extracts using Dsn1-Flag (Akiyoshi et al, 2010). Using a proximity-dependent in vivo-biotinylation approach with the TurboID Ligase (Fenech et al,

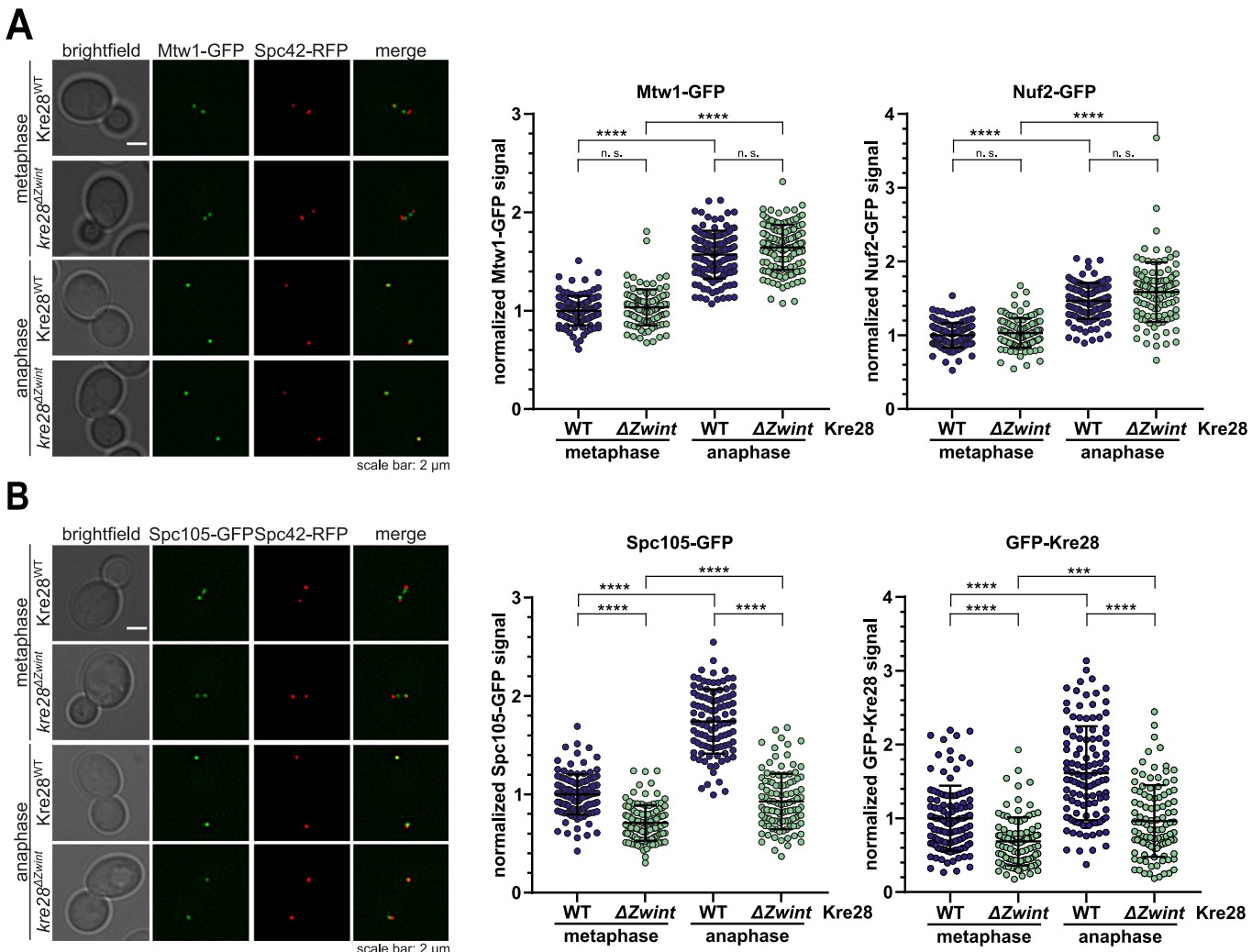

**Figure 3. The Kre28^ΔZwint mutant reduces the level of Spc105/Kre28 at kinetochore clusters in vivo.**

(A) Live-cell imaging of yeast cells with GFP-tagged Mtw1 and Nuf2 subunits. Representative images of metaphase and anaphase cells are shown on the left, quantification of fluorescence signal intensities on the right. Kinetochore recruitment of Mtw1-GFP (Mtw1 complex), Nuf2-GFP (Ndc80 complex) was analyzed in Kre28^WT and kre28^ΔZwint strain backgrounds. The C-terminus of Spc42 was fused to RFP to label the position of the mitotic spindle. The mean of metaphase signal intensities in the wild-type strain was defined as 1 and all other data was normalized to this value. Means +/− standard deviation are plotted. P values were calculated with a Kruskal–Wallis test and are displayed as follows: n.s. nonsignificant = P value > 0.05; ****P < 0.0001. Exact P values from (A): n.s. >0.9999 in all cases, except Mtw-GFP anaphase wt versus Δzwint P = 0.5820; ****P < 0.0001 in all cases. n > 100 kinetochore clusters were analyzed for each condition. Representative data from at least two independent experiments is shown. (B) Live-cell imaging of yeast cells with GFP-tagged Spc105 and GFP-Kre28. Representative images of metaphase and anaphase cells are shown on the left, and the quantification of fluorescence signal intensities on the right. Kinetochore recruitment of Spc105 was analyzed in Kre28^WT and kre28^ΔZwint strain backgrounds. GFP-Kre28 was analyzed as wild-type and Δzwint mutant. The C-terminus of Spc42 was fused to RFP to label the position of the mitotic spindle. The mean of metaphase signal intensities in the wild-type strain was defined as 1, and all other data were normalized to this value. Means +/− standard deviation are plotted. P values were calculated with a Kruskal–Wallis test and displayed as follows: n.s. nonsignificant = P value > 0.05; ****P < 0.0001; ***P < 0.001; Exact P values from (B): ****P < 0.0001 in all cases; ***P = 0.0003. n > 100 kinetochore clusters were analyzed for each condition. Representative data from at least two independent experiments is shown. Source data are available online for this figure.

2023) we found, however, that Sli15-6xFlag could be detected in Streptavidin affinity-purifications from extracts of strains that expressed Spc105-TurboID, similar to the constitutive Spc105 binding partner Kre28 (Fig. 5F). The nuclear kinesin-5 motor protein Cin8, expressed at a lower level and implicated in the regulation of kinetochores via Ndc80 (Gardner et al, 2008; Suzuki et al, 2018), on the other hand, was not recovered with Spc105-TurboID. This supports the idea that Spc105-Sli15 interactions also occur in vivo, although possibly only transiently.

To characterize the KMN-Ipl1/Sli15 assembly in more detail, we conducted chemical cross-linking with the homo-bifunctional amine-to-amine crosslinker BS3, followed by mass spectrometry (XL-MS) (Herzog et al, 2012). The resulting Lys-Lys proximity map is in overall agreement with the established KMN subunit topology (Fig. 6A,B). In particular, it showed that the C-termini of the subunits Mtw1, Dsn1, and Nsl1 are closely connected to each other and proximal to Spc105/Kre28 and to the Spc24/Spc25 heterodimer of the Ndc80 complex (Dimitrova et al, 2016; Ghodgaonkar-Steger et al, 2020; Malvezzi et al, 2013). The

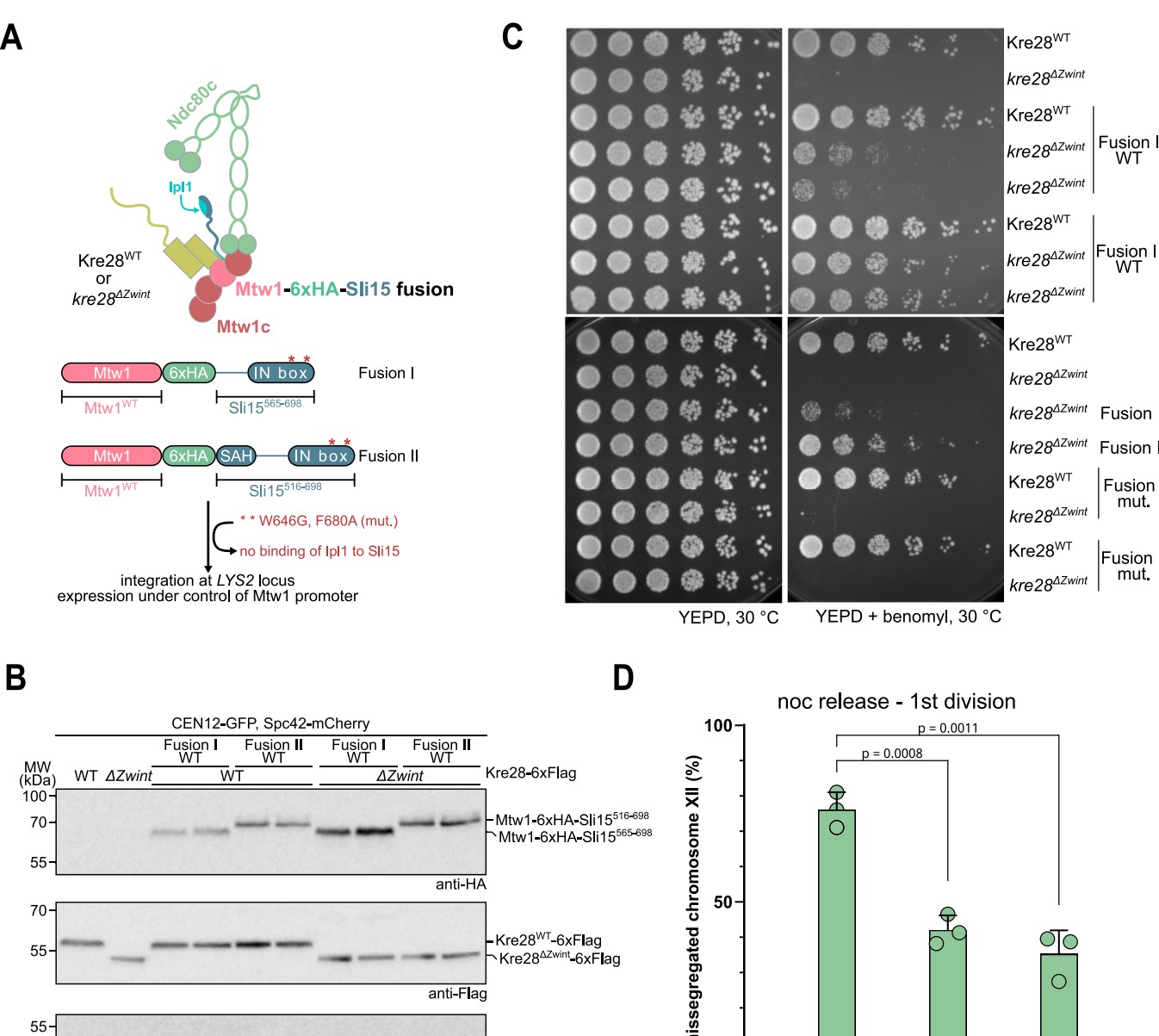

**Figure 4. Artificial recruitment of Sli15 to Mtw1 suppresses the _kre28^ΔZwint_ mutant phenotype.**

(**A**) Illustration of artificial recruitment of Ipl1 to KMN by fusion of a C-terminal fragment of Sli15 to the C-terminus of Mtw1 and schematic of the architecture of two different Mtw1-Sli15 fusion proteins. Two C-terminal fragments of Sli15 were fused to the C-terminus of Mtw1, separated by a 6xHA tag. Ipl1 is recruited to the fusion proteins by binding to the IN box of Sli15. The red asterisks mark the position of W646G and F680A amino acid substitutions which prevent binding of Ipl1 to Sli15. The genes encoding the fusion constructs were integrated at the _LYS2_ locus and under control of the Mtw1 promotor. (**B**) Western blot analysis of Mtw1 fusion proteins and Kre28 mutants. Extracts of indicated strains were blotted for the presence of fusion proteins (anti-HA), Kre28 wild-type or mutant (anti-Flag). (**C**) Serial dilution assay with haploid yeast strains with Kre28^WT or _kre28^ΔZwint_ expressing different Mtw1-Sli15 fusion proteins. Serial dilutions of yeast strains with the indicated genotypes were spotted on YEPD and YEPD + benomyl plates and incubated at the indicated temperatures. (**D**) Scoring of chromosome segregation in Kre28^WT or _kre28^ΔZwint_ with fluorescently labeled chromosome XII (CEN12-GFP) and spindle pole bodies (Spc42-mCherry) expressing different Mtw1-Sli15 fusion proteins. Cells were released from a nocodazole arrest, and segregation of chromosome XII in the first anaphase after release was analyzed. Missegregation rates in the different strains are displayed with data from three independent experiments. Mean values +/− standard deviation are plotted. The dots represent the data from three individual experiments. _n_ > 110 cells were analyzed for each condition in each experiment. _P_ values were calculated with an unpaired _t_ test. Source data are available online for this figure.

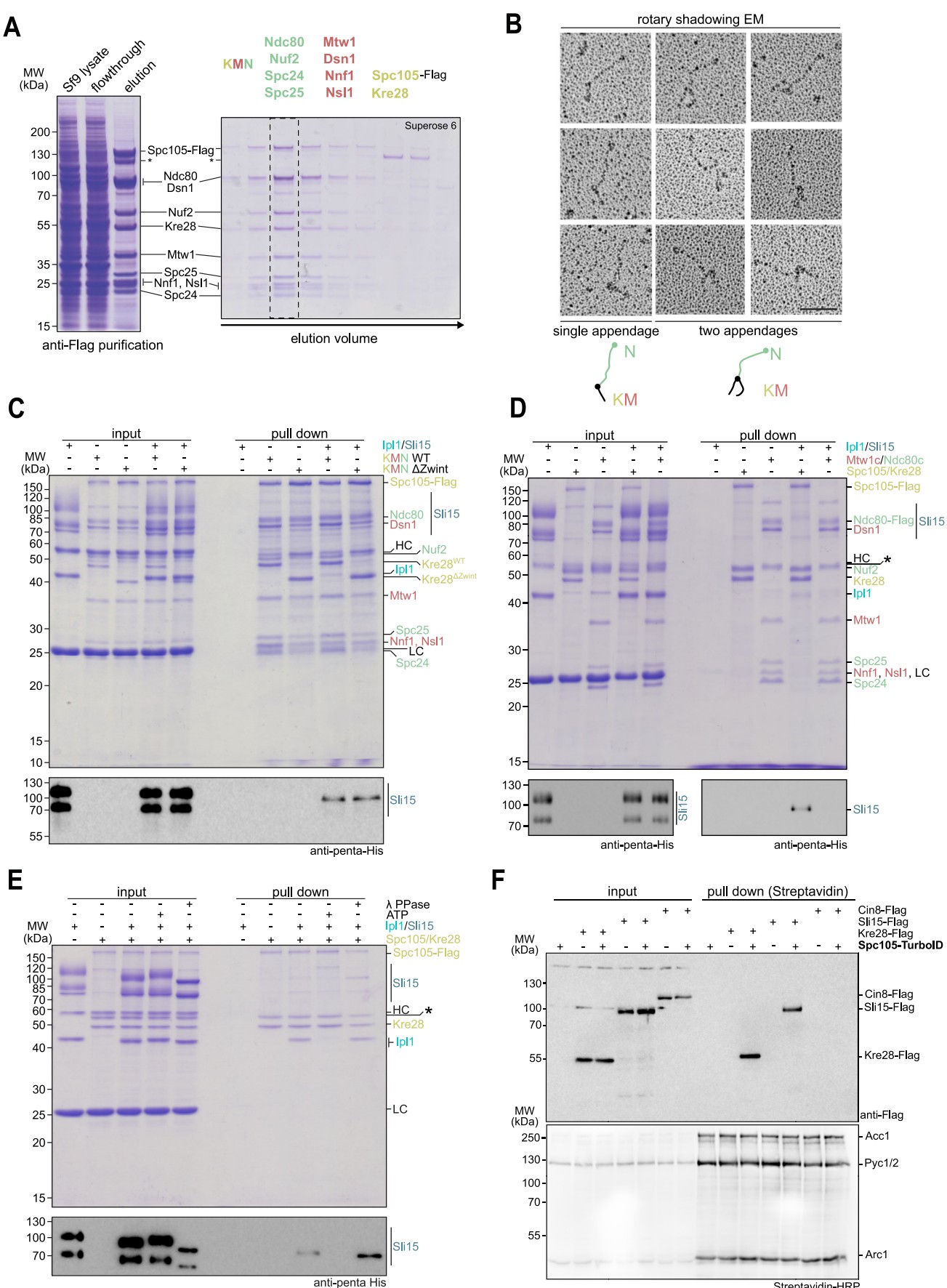

**Figure 5.   Ipl1/Sli15 binds to KMN via the Spc105/Kre28 subcomplex in vitro.**

(A) Single-step isolation of a recombinant budding yeast KMN network from Sf9 insect cells. Left side shows Coomassie-stained SDS-PAGE of anti-Flag purification of Spc105-Flag. Right panel shows consecutive fractions from size exclusion chromatography on a Superose 6 column. (B) Gallery of electron micrographs showing recombinant KMN network visualized by Pt/C rotary shadowing EM. Distinct configurations are shown in which the end of the Ndc80 complex shows one or more appendages. Scale bar: 50 nm. (C) Pull-down assays to test for binding of Ipl1/Sli15 to KMN. KMN containing either Kre28$^{WT}$ or Kre28$^{\Delta Zwint}$ was purified from Sf9 insect cells and immobilized on M2 anti-Flag beads. Recombinant Ipl1/Sli15 was added to test for its binding to KMN. Proteins were eluted from beads by the addition of 3xFlag peptide. Input and pull-down samples were analyzed by SDS-PAGE and Coomassie staining or western blot to detect 6xHis-tagged Sli15. (D) Pull-down assay to test for binding of Ipl1/Sli15 to Spc105/Kre28 or Mtw1c/Ndc80c. Either Spc105/Kre28 or Mtw1c in combination with Ndc80c were purified from Sf9 insect cells and immobilized on M2 anti-Flag beads. Recombinant Ipl1/Sli15 was added to test for its binding to the immobilized proteins. Proteins bound to the beads were finally eluted with 3xFlag peptide and input and pull-down samples were analyzed by SDS-PAGE followed by Coomassie staining or western blot to detect 6xHis-tagged Sli15. Asterisk denotes potential proteolysis product of full-length Spc105. (E) Pull-down assay to analyze the impact of Ipl1/Sli15 phosphorylation on binding to Spc105/Kre28. Spc105/Kre28 was purified from Sf9 insect cells and immobilized on M2 anti-Flag beads. Prior to addition to the loaded beads, recombinant Ipl1/Sli15 was in vitro autophosphorylated (+ATP), dephosphorylated by λ phosphatase, or left untreated. After incubation with the immobilized Spc105/Kre28, bound proteins were eluted from the beads using 3xFlag peptide and input and pull-down samples were analyzed by SDS-PAGE and Coomassie staining or western blot to detect 6xHis-Sli15. The mobility of Ipl1 and Sli15 during SDS-PAGE depends on their phosphorylation status. LC and HC mark the light and heavy chains of the M2 anti-Flag antibody. The asterisk denotes the potential proteolysis product of full-length Spc105. (F) In vivo proximity-biotinylation experiment using Spc105-TurboID and different Flag-tagged potential interactors. Yeast lysates were prepared from the indicated strains and subjected to a Streptavidin affinity purification to isolate biotinylated proteins. Note that Kre28 and Sli15 are only detected in pull downs from strains that also express Spc105-TurboID. The major endogenously biotinylated proteins Acc1, Pyc1/2, or Arc1 enriched in the Streptavidin pulldown are indicated. Source data are available online for this figure.

majority of Sli15/Ipl1 crosslinks to the KMN network is found on the Spc105-Kre28 subcomplex (17 different crosslinks to Spc105/Kre28, 4 to Mtw1c, and 4 to Ndc80c). In particular, the Spc105 segment 450–700, encompassing the helix bundle with Kre28, is found in proximity to Sli15. On Sli15 most crosslinks are found between residues 200 and 250, marking the transition between the amino-terminal CEN localization module and the central phospho-regulated (PR) domain, previously implicated in microtubule-binding. For instance, Sli15 K217 displays crosslinks to Spc105 K527 and K607, both located in the helix bundle domain. Overall, these crosslinks are consistent with the binding experiments, showing that Spc105/Kre28 is the main interaction partner of Ipl1/Sli15 in the outer kinetochore KMN network.

## Binding of a Sli15 segment to the Spc105/Kre28 helix bundle is essential for chromosome segregation in cells

Guided by the cross-linking results, we performed AlphaFold structure predictions with N-terminal Sli15 segments of varying length. Strikingly, Sli15 segment 215–255 yielded a high confidence model on the Spc105/Kre28 helix bundle (pTM and ipTM scores >0.85) (Figs. 7A and EV4A,B). The predicted binding mode includes a short alpha-helix at the N-terminus of the Sli15 peptide (residues 224–236) and an extended segment that binds across the surface of the helix bundle (Fig. 7A). The confidence of the prediction is highest in the extended segment (236–248), while the alpha-helix shows some positional variance between models (Fig. EV4B). The Kre28 Zwint helix contributes only its C-terminus to the predicted binding interface, the segment which also yielded the most severe phenotypes in our mutational analysis (Figs. EV4C and EV1). The Sli15 arginine pair R231 and R232 is predicted to form ionic interactions with Spc105 E510 and Spc105 E596. The Sli15 residue L239 is buried in a hydrophobic pocket formed by Spc105 residues W579, W582 and L519, and Sli15 K242 interacts with an acidic pocket formed by Spc105 E578, Kre28 E103, and Kre28 D106 (Fig. 7B). The crucial residues are part of a conserved segment among yeast Sli15 proteins (Fig. 7C). We generated a $sli15^{4E}$ mutant exchanging these residues. An in vitro pull-down experiment showed reduced binding of Ipl1/6xHis-Sli15$^{4E}$ to immobilized Spc105-Kre28 (Fig. 7D). We next replaced one copy of $SLI15$ with Flag-tagged wildtype or $sli15^{4E}$ in diploid yeast cells. We constructed the

4E mutation in full-length Sli15 and in the Sli15$^{\Delta N}$ variant lacking the CEN localization module. Upon sporulation and dissection, we failed to recover viable $sli15^{4E}$ mutant spores, both in full-length and in ΔN form (Fig. 7E). This indicates that $sli15^{4E}$ is a lethal mutation, despite the mutant protein being expressed at the same level as wild type in the diploid (Appendix Fig. S3A,B). Confirming this result, $sli15^{4E}$ also failed to rescue the acute depletion of $sli15$-aid in haploid yeast strains using an auxin-dependent degron system (Fig. EV5A–C). Dissecting the contributions of individual residues in the auxin-degron system showed that $sli15^{R231E\ R232E}$ was viable, but benomyl-hypersensitive, while $sli15^{K242E}$ displayed a severe growth defect, similar to the 4E mutant (Fig. EV5D). The AlphaFold model also predicts that corresponding mutations in Spc105 should yield similar growth phenotypes. Consistent with this notion, the charge-reversing mutations $spc105^{E578K\ E596K}$ displayed benomyl hypersensitivity and $spc105^{E510K\ E578K\ E596K}$ had a severe growth defect in a conditional Spc105-FRB anchor-away system (Fig. EV5E,F).

To analyze differences between wild type and mutant $sli15^{4E}$ in cells, we imaged mNeonGreen-tagged variants in diploid yeast cells together with a kinetochore marker (Mtw1-mCherry). While we did not observe significant localization differences between the full-length Sli15 versions, we noticed that $sli15^{\Delta N}$, unlike the full-length protein, localized very prominently to the ends of anaphase spindles, displaying enrichment and precise overlap with the kinetochore marker Mtw1 (Fig. 7F). Importantly, the fraction of anaphase spindles showing co-enrichment of Mtw1 and $sli15^{\Delta N,4E}$ was significantly reduced (Fig. 7G). The 4E mutation did not affect the overall spindle association and the difference to wild type was most notable on anaphase kinetochores. We conclude that the 4E mutation impairs Sli15 binding to a subset of kinetochores in cells, a defect that is revealed when the CEN localization module of Sli15 is absent.

# Discussion

## New $kre28$ mutant alleles reveal specific contributions to outer kinetochore function

While the molecular roles of Ndc80c and Mtw1c complexes in the kinetochore architecture are well-established, the functional

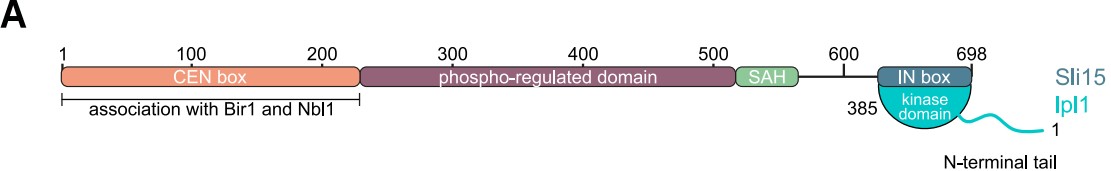

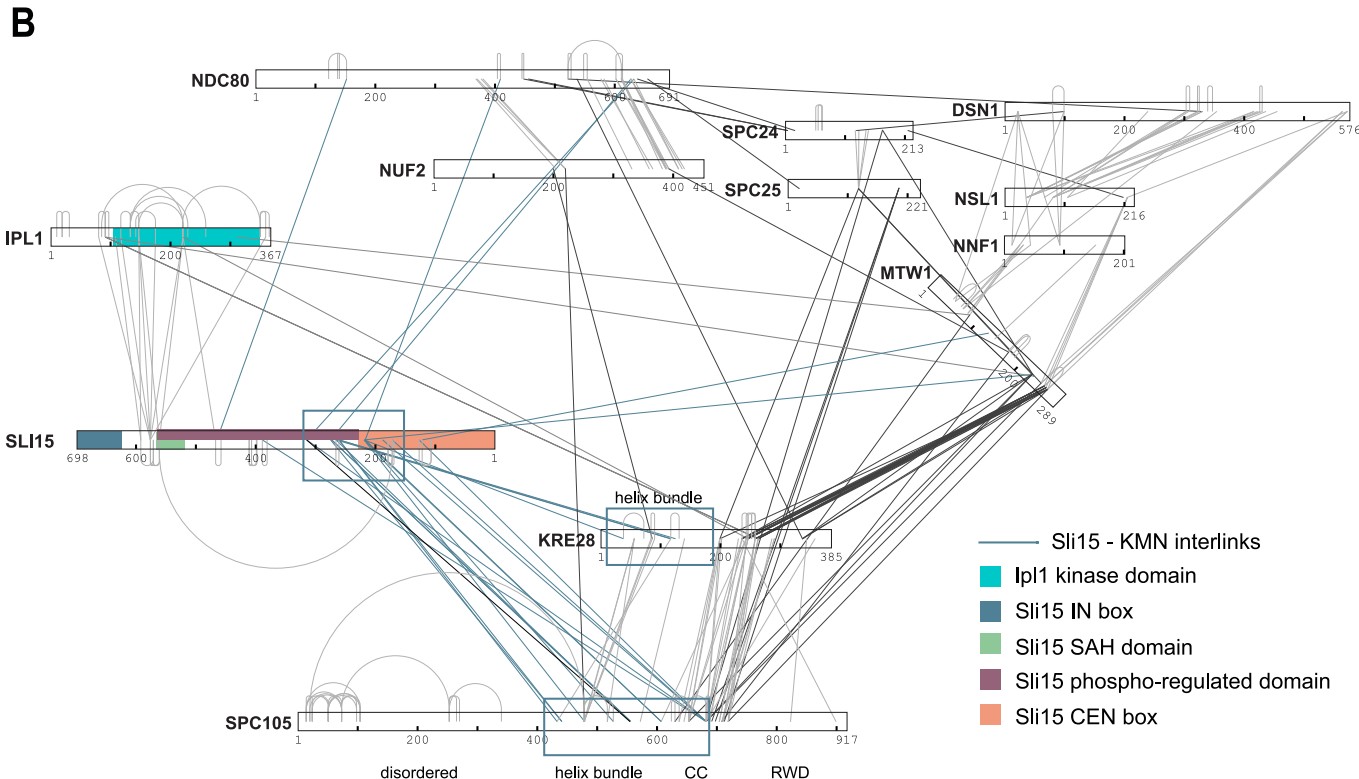

**Figure 6. Chemical cross-linking mass spectrometry analysis of the KMN-Ipl1/Sli15 assembly.**

(A) Schematic illustration of Sli15 and Ipl1. Sli15 associates with its N-terminal CEN box to Bir1/Survivin and Nbl1/Borealin. The C-terminal IN box binds to the globular kinase domain of Ipl1. (B) Network diagram of inter- and intramolecular Lys-Lys crosslinks of a KMN-Ipl1/Sli15 assembly. Recombinant KMN and Ipl1/Sli15 were mixed at equimolar amounts and incubated in the presence of BS3. Functional domains of Ipl1 and Sli15 are indicated in the legend. Interlinks originating from Sli15 to KMN subunits are highlighted. Segments rich in crosslinks between Sli15 and Spc105 are boxed.

contributions of Spc105/Kre28 still await further definition. The mitotic checkpoint function of Spc105 is well understood, but it cannot account for its essential role in budding yeast. The *kre28*$^{\Delta Zwint}$ allele described in this study can be used to reveal additional aspects of Spc105/Kre28 function. We show that one of its effects is a selective reduction of Spc105/Kre28 at the kinetochore without affecting the levels of Ndc80 and Mtw1 complexes. It is therefore much more selective than more penetrant perturbations. For example, a depletion or reduction of human Knl1 by RNAi strongly affects the levels of additional outer kinetochore components such as Ndc80/Hec1 as well (Caldas et al, 2013; Desai et al, 2003), making it difficult to delineate a specific function. We note that the observed Spc105/Kre28 reduction in metaphase cells is moderate and does not lead to impaired spindle assembly checkpoint activity, as the mutant cells are fully capable of arresting in nocodazole. In addition, they show normal levels of Sgo1 recruitment to the kinetochore in prometaphase, suggesting sufficient Bub1 activity at the kinetochore. We currently cannot provide a precise molecular explanation for why the mutations in the Kre28 helical bundle may reduce the affinity of Spc105/Kre28 for Ndc80c/Mtw1c, both in vivo and in vitro. Even though

AlphaFold2 models predict multiple possible conformations of KMN, they do not show a binding interface within KMN that would include the Kre28 Zwint helix. It is also possible that diminished kinetochore recruitment may be a consequence of the reduced Ipl1 activity in the mutant. Future experiments will have to clarify this point using some of the more specific mutations discovered in our study.

## An outer kinetochore recruitment mechanism for the Ipl1/Sli15 kinase complex

Taken together, the following lines of evidence support the conclusion that the *kre28* helical bundle mutant phenotype is caused by insufficient or misregulated Ipl1 activity at the outer kinetochore.

1. The phenotypes of the *kre28*$^{\Delta Zwint}$ mutant strongly resemble defects in the tension-dependent error correction pathway. The mutant cells have a functional mitotic checkpoint in response to nocodazole, but when forced to reassemble a spindle following nocodazole withdrawal, they ignore biorientation problems and missegregate

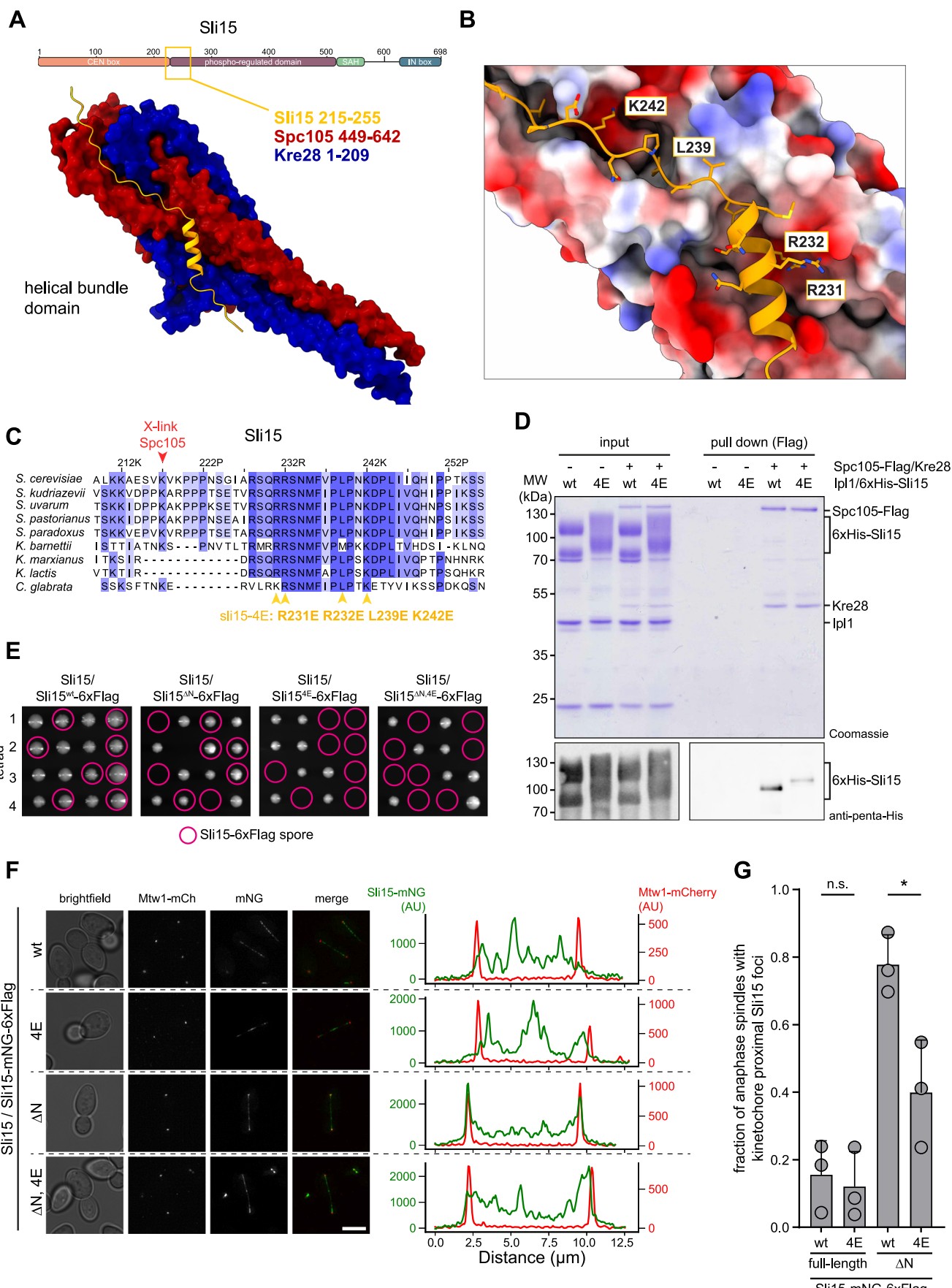

◀

**Figure 7.  Binding of a Sli15 segment to the Spc105/Kre28 helical bundle is critical for chromosome segregation in cells.**

(A) Alphafold2 structure prediction of the Spc105/Kre28 helical bundle domain (dark red/dark blue) bound to Sli15 segment 215–255 (orange). Scheme on top indicates the position of the peptide in the Sli15 domain organization. (B) Details of the predicted Sli15-binding interface. The surface of Spc105/Kre28 is indicated as electrostatic potential (red: negatively charged, blue positively charged, white: neutral). Predicted key Sli15 residues and their side chains are highlighted. (C) Multiple sequence alignment of the Sli15 segment from different yeast species. MSA was generated in Jalview and colored based on conservation. The position of residues mutated in the *sli15^4E* mutant is indicated. *S. Saccharomyces*, *K. Kluyveromyces*, *C. Candida*. The proximal Sli15 residue K217 was found to crosslink to Spc105 (Fig. 5). (D) Pull-down assay with recombinant Spc105/Kre28 and Ipl1/6xHis-Sli15 wild-type or Sli15^4E mutant. After incubation with the immobilized Spc105/Kre28, bound proteins were eluted from the beads using 3xFlag peptide, and input and pull-down samples were analyzed by SDS-PAGE and Coomassie staining or western blot to detect 6xHis-Sli15. (E) Representative images of dissection plates from diploid Sli15/Sli15-6xFlag strains, in which the Sli15-6xFlag allele encodes for either Sli15^wt, sli15^4E, sli15^ΔN, or sli15^ΔN, 4E. Circled spores indicate the recovered or inferred Sli15-6xFlag-tagged variants. (F) Left: representative micrographs of diploid cells in late anaphase. The cells are heterozygous for Mtw1/Mtw1-mCherry and Sli15/Sli15-mNeonGreen(mNG)-6xFlag. The integrated locus of Sli15 encodes for either Sli15^wt, sli15^4E, sli15^ΔN, or sli15^ΔN, 4E. Scale bar: 2 μm. Right: line scan analysis of the intensity of Mtw1-mCherry and Sli15-mNeonGreen along the anaphase spindle in the cells depicted on the left. Only Sli15^ΔN shows sharp peaks for Sli15-mCherry that co-localize with the peaks for Mtw1-mCherry. (G) Quantification of the fraction of anaphase spindles that show Sli15-mNeonGreen foci at spindle poles, overlapping with the Mtw1-mCherry foci, as represented in (F). The points indicate the fraction in three replicates, the bar depicts the mean with standard deviation. Statistical analysis: unpaired *t* test comparing between wild type and 4E. n.s. nonsignificant $P > 0.05$, *$P < 0.05$. Exact $P$ values for (G) n.s. $P = 0.6972$, *$P = 0.0219$. Source data are available online for this figure.

chromosomes without delaying the initiation of anaphase. The lack of a delay indicates that the cells can form kinetochore–microtubule attachments, because otherwise the unattached kinetochores would activate the checkpoint. This characteristic behavior is also displayed by *sgo1* mutants (Indjeian et al, 2005) and indicates an inability to sense the incorrect attachment status.

2. The genetic interaction profile indicates that the *kre28* mutant is very sensitive to further perturbations of Ipl1/Sli15, its recruitment pathways, or mutations in its outer kinetochore substrates.

3. The *kre28^ΔZwint* mutant is suppressed by the artificial recruitment of Ipl1 via an Mtw1-Sli15 fusion protein. We show that the Mtw1-Sli15 fusion can only suppress the phenotype when capable of binding Ipl1. The observed rescue is only partial, partly due to the fact the fusion protein is expressed over a wild-type Mtw1 and therefore will have to compete with it for kinetochore incorporation. The fusion is also constitutive, preventing a dynamic regulation of Ipl1 recruitment that may occur in wild-type cells. We nevertheless suspect that fusing the Sli15 segment to the C-terminus of Mtw1 reintroduces Ipl1 at a compatible location within KMN, while previous studies have reported constitutive Sli15 fusions to Ndc80 to be incompatible with biorientation (Li et al, 2023). Our live-cell imaging experiment using fluorescently labeled chromosomes demonstrates that improved chromosome segregation after nocodazole washout is the basis for the suppression of the growth defect.

4. Recombinant Ipl1/Sli15 binds KMN via the Spc105/Kre28 subunits in vitro. The chemical cross-linking-mass spectrometry experiments are consistent with Spc105/Kre28 providing the main binding site within the KMN network. Autophosphorylation of Ipl1/Sli15 inhibits binding, possibly preventing excessive Ipl1/Sli15 recruitment, which would hinder the re-establishment of kinetochore–microtubule attachments during error correction. Also, phosphorylation of Spc105/Kre28 may contribute to regulatory mechanisms, this point will need to be dissected in futures studies. Our findings are consistent with earlier reports showing that depletion of Knl1 appears to reduce Aurora B substrate phosphorylation in human cells (Caldas et al, 2013).

5. A Sli15 peptide motif in the central domain is predicted with high confidence to bind the Spc105/Kre28 helical bundle. AlphaFold-guided mutations in Sli15 or corresponding mutations in Spc105/Kre28 are lethal or have severe growth defects, consistent with fatal problems in chromosome segregation.

## Implications for the tension-sensing mechanism of the kinetochore

Our results re-affirm the notion that Bir1/Nbl1-based CEN recruitment of the CPC is not required for chromosome biorientation in yeast cells (Fig. 8A). Previously, this has been attributed to enhanced clustering of the Sli15^ΔN complex on microtubules (Campbell and Desai, 2013). Our results allow a more straightforward explanation: The Sli15^ΔN complex contains crucial outer kinetochore binding sites that are sufficient to provide full functionality during chromosome biorientation. We show that a segment in the Sli15 central domain binds to the Spc105/Kre28 helix bundle. Mutational analysis indicates that this binding interface is a key element in the tension-dependent error correction mechanism. Crucially, the *sli15^4E* mutant does not only cause a growth phenotype in the *sli15^ΔN* version, but also in the full-length protein. Binding of Sli15 to Spc105/Kre28 is therefore not just a backup mechanism that only becomes important when the regular CEN-based localization is absent, but it must be an important element of the tension-sensing and signaling mechanism in general. The lethality of the *sli15^4E* point mutant implies that tension-sensing must occur in the outer kinetochore KMN architecture. Defining the terminal phenotype of the *sli15^4E* mutant, more precisely, will be an important point in future studies.

Previous models have emphasized the role of CPC centromere-anchoring in separating the kinase from its substrates under tension (Liu et al, 2009). Sli15^INCENP performs a key function as it can restrict the reach of the kinase in the kinetochore (Krenn and Musacchio, 2015). Our results reveal that Sli15 needs to be connected to very specific points within the KMN architecture in order to measure intra-kinetochore tension (Fig. 8B). Notably, movements of the K-arm (including Spc105/Kre28) relative to other parts of the kinetochore have been documented (Wan et al, 2009). The Ipl1/Sli15 kinase module of the CPC is therefore an integral part of the outer kinetochore and can be influenced by architectural and conformational changes that occur in different attachment situations. Future experiments will have to fully define the underlying outer kinetochore binding mechanism and ask how the Ipl1 kinase activity towards key substrates may be influenced by distinct KMN conformations under force.

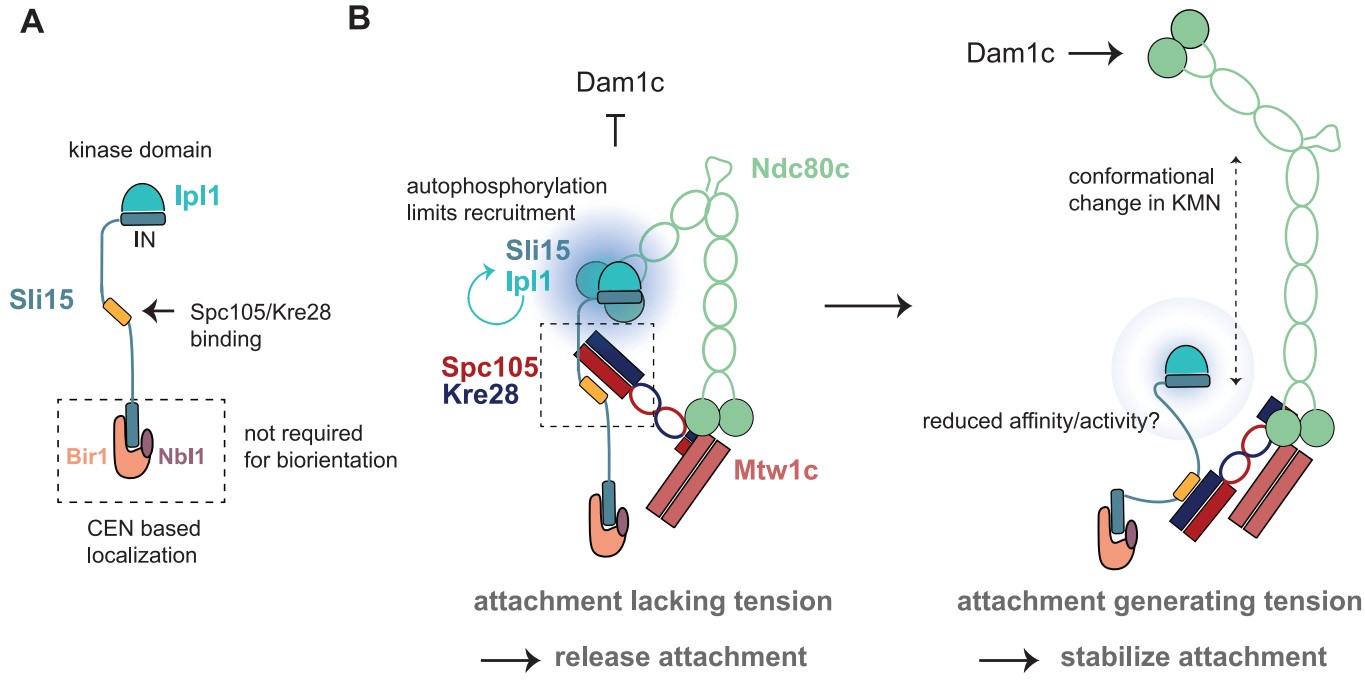

**Figure 8. A model for Ipl1/Sli15 docking to the outer kinetochore.**

(A) Organization of centromere-binding modules and outer kinetochore docking sites in the Ipl1/Sli15 complex. (B) Model for how conformation-sensitive outer kinetochore docking could contribute to tension-dependent error correction. To improve clarity, the disordered N-terminus of Spc105 has been omitted. A crucial interface between a Sli15 segment and the Spc105/Kre28 helix bundle is highlighted.

# Methods

### Reagents and tools table

| Reagent/resource | Reference or source | Identifier or catalog number |
|---|---|---|
| **Experimental models** | | |
| BL21 (DE3) *E. coli* | Thermo Fischer | Cat. #EC0114 |
| BL21 Rosetta (DE3) *E. coli* | Fischer Scientific | Cat. #70-954-4 |
| High Five insect cells | Thermo Fischer | Cat. #10712024 |
| Sf9 insect cells | Thermo Fischer | Cat. #11496015 |
| *S.cerevisiae* S288C | Appendix Table S1 | This study |
| **Recombinant DNA** | | |
| pAD77 | Kre28$^{WT}$-6xFlag in pRS306 | This study |
| pAD78 | Kre28$^{\Delta Zwint}$-6xFlag in pRS306 | This study |
| pJP1 | Kre28$^{\Delta 77\text{-}90}$-6xFlag in pRS306 | This study |
| pJP2 | Kre28$^{\Delta 91\text{-}102}$-6xFlag in pRS306 | This study |
| pJP3 | Kre28$^{\Delta 103\text{-}113}$-6xFlag in pRS306 | This study |
| pAD81 | Kre28$^{4A}$-6xFlag (L95A, K96A, Y99A, E101A) in pRS306 | This study |
| pAD82 | Kre28$^{3A}$-6xFlag (E103A, L105A, D106A) in pRS306 | This study |

| Reagent/resource | Reference or source | Identifier or catalog number |
|---|---|---|
| pAD84 | Kre28$^{6A}$-6xFlag (F108A, F109A, R110A, F111A, T112A, L113A) in pRS306 | This study |
| pAD89 | Kre28$^{WT}$-GFP in pRS306 | This study |
| pAD90 | Kre28$^{\Delta Zwint}$-GFP in pRS306 | This study |
| pAD98 | Mtw1-6xHA-Sli15$^{565\text{-}698}$ in pRS307 | This study |
| pAD101 | GFP-Kre28$^{WT}$-6xFlag in pRS306 | This study |
| pAD102 | GFP-Kre28$^{\Delta Zwint}$-6xFlag in pRS306 | This study |
| pAD103 | Kre28$^{\Delta RWD}$-6xFlag in pRS306 | This study |
| pAD104 | Kre28$^{\Delta Zwint\ \Delta RWD}$-6xFlag in pRS306 | This study |
| pAD105 | Mtw1-6xHA-Sli15$^{565\text{-}698}$ (W646G, F680A) in pRS307 | This study |
| pAD108 | Mtw1-6xHA-Sli15$^{516\text{-}698}$ in pRS307 | This study |
| pAD118 | Mtw1-6xHA-Sli15$^{516\text{-}698}$ (W646G, F680A) in pRS307 | This study |
| pAD120 | Mtw1-6xHA-Sli15$^{565\text{-}698}$::natNT2 in pRS307 | This study |
| pAD121 | Mtw1-6xHA-Sli15$^{516\text{-}698}$::natNT2 in pRS307 | This study |
| pAD112 | Sli15-6xFlag in pRS303 | This study |
| pAD119 | Sli15 D2-228-6xFlag in pRS303 | This study |
| pAD144 | Sli15 R231E R232E L239E K242E -6xFlag in pRS303 | This study |

| Reagent/resource | Reference or source | Identifier or catalog number |
|---|---|---|
| pAD145 | Sli15 deltaN R231E R232E L239E K242E -6xFlag in pRS303 | This study |
| pAD146 | Sli15 R231E R232E L239E K242E in pETDuett | This study |
| pAD147 | Sli15-mNeonGreen-6xFlag in pRS303 | This study |
| pAD148 | Sli15 deltaN-mNeonGreen-6xFlag in pRS303 | This study |
| pAD149 | Sli15 R231E R232E L239E K242E -nNG-6xFlag in pRS303 | This study |
| pAD150 | Sli15 deltaN R231E R232E L239E K242E-mNG -6xFlag in pRS303 | This study |
| pAD153 | Sli15-6xFlag in pRS305 | This study |
| pAD154 | Sli15 delta2-228-6xFlag integration construct in pRS305 | This study |
| pAD155 | Sli15 R231E R232E L239E K242E -6xFlag in pRS305 | This study |
| pAD156 | Sli15 deltaN R231E R232E L239E K242E in pRS305 | This study |
| pAD158 | Spc105 E510K E578K E596K in pRS306 | This study |
| pAD160 | Sli15 R231E R232E -6xFlag (integration construct) in pRS305 | This study |
| pAD161 | Sli15 K242E -6xFlag in pRS305 | This study |
| pFAP10_Baculovirus | Spc105-Flag, Kre28$^{WT*}$ | This study |
| pFAP10 mut_Baculovirus | Spc105-Flag, Kre28$^{\Delta Zwint}$ | This study |
| KT1-5_Baculovirus | Dsn1, Mtw1, Nnf1, Nsl1, Spc105-Flag | This study |
| KT1-6_Baculovirus | Ndc80, Nuf2, Spc24, Spc25, Kre28$^{WT}$ | This study |
| KT1-6 _Baculovirus mut | Ndc80, Nuf2, Spc24, Spc25, Kre28$^{\Delta Zwint}$ | This study |
| Ndc80c-Flag_Baculovirus | Ndc80-Flag, Nuf2, Spc24, Spc25 | Pleuger et al, 2024 |
| pFAP19_Baculovirus | Dsn1, Mtw1, Nnf1, Nsl1 | This study |
| **Antibodies** | | |
| Anti-Flag (HRP-conjugated) (M2) | Sigma-Aldrich | A8592 |
| Anti-HA | Biolegend | 901514 |
| Anti-Myc (9E10) | Santa Cruz Biotechnology, Inc | sc-40 |
| Anti-GFP | Roche | 11814460001 |
| Anti-Pgk1 (22C5D8) | Invitrogen | 459250 |
| Anti-penta-His (HRP-conjugated) | Qiagen | 34460 |
| Anti-mouse IgG (HRP-conjugated) | Cytiva | NA931V |
| **Chemicals, enzymes, and other reagents** | | |
| Anti-Flag M2 affinity gel | Sigma-Aldrich | SKU A2220 |
| cOmplete protease inhibitor | Roche | SKU 11697498001 |

| Reagent/resource | Reference or source | Identifier or catalog number |
|---|---|---|
| FLAG peptide | CASLO ApS | On demand |
| HisTrap HP | Cytiva | 17524802 |
| Naphtol acetic acid NAA | Roth | #4343.1 |
| NiNTA agarose | Takara | Cat. #635660 |
| PhosStop phosphatase inhibitor | Roche | SKU 4906845001 |
| **Software** | | |
| FIJI/ImageJ | ImageJ | RRID:SCR_002285 |
| SoftWoRx 6 | GE Healthcare | RRID:SCR_019157 |
| **Other** | | |
| μ-Slide 8 Well high Glass Bottom | Ibidi | Cat. #80806 |

## Cultivation and genetic manipulation of yeast

All yeast strains used in this study are based on the S288C strain. Standard techniques were used for the cultivation and genetic manipulation of yeast strains. PCR-generated integration cassettes were generated according to standard techniques. Alternatively, linearized plasmids with ends homologous to the site of integration were used for transformation. Plasmids used for yeast transformation are listed in the reagents and tools table, a full list of yeast strains it provided in Appendix Table S1.

## Sporulation and dissection of diploid yeast strains

Cells from 1 ml of a saturated overnight culture of diploid yeast strains were washed twice with 1 ml ddH$_2$O, resuspended in 4 ml sporulation medium and incubated at room temperature for several days until tetrads were formed. 250 μl of the sporulation culture were centrifuged and the cell pellet was resuspended in 37.5 μl Zymolase T (1 mg/ml) and incubated at 37 °C for 3.5 min. Afterward, 200 μl of cold 100 mM potassium phosphate buffer (pH 7.0) was added. A few microliters of the cell suspension were streaked out on a YEPD plate, and tetrads were dissected using a Singer MSM semi-automated dissecting microscope. Plates were incubated at 30 °C for 3 days. For genotyping, the spores were resuspended in minimal medium and spotted on YEPD, drop-out and mating-type tester plates.

## Serial dilution assay

Cells from a saturated overnight culture were diluted to OD$_{600}$ = 0.4 in minimal medium and serially diluted (1:4, six dilution steps in total). Cells were spotted on YEPD or YEPD + benomyl (20 μg/ml; Sigma-Aldrich) plates using a pinning tool. Plates were incubated at 25 °C, 30 °C, or 37 °C for 2 days or at 16 °C for 5 days.

## Preparation of western blot samples from yeast cultures

Samples for western blot analysis were prepared according to a protocol adapted from (Kushnirov, 2000). In brief, an equivalent of 2

$OD_{600}$ from a logarithmically growing culture was harvested by centrifugation, resuspended in 100 µl ddH$_2$O and 100 µl 0.2 M NaOH was added. Samples were incubated at room temperature for 5 min, cells were pelleted, resuspended in 50 µl SDS-PAGE loading buffer and boiled at 95 °C for 5 min. In total, 6–8 µl of sample were loaded for SDS-PAGE. Proteins were transferred to a Amersham™ Protran™ Nitrocellulose Blotting Membrane (Cytiva) by wet transfer. Membranes were blocked in TBST with 5% (w/v) skimmed milk, incubated with the respective antibodies (listed in R&T table) and membranes were washed in TBST. For the detection of signals from HRP-conjugated antibodies, membranes were incubated with ECL™ (Prime) Western Blotting Detection Reagents (Cytiva). Signals were visualized using an Amersham Imager 600 (GE Healthcare).

## Analysis of cell cycle progression after release from α factor arrest into medium with nocodazole

Cells from a saturated overnight culture were diluted to $OD_{600} = 0.2$ and incubated at 30 °C for 1 h. α factor was added to a final concentration of 10 µg/ml, and cells were further incubated at 30 °C for 2–2.5 h. Arrest in G1 phase was confirmed by microscopy when more than 90% of cells had a shmoo. Cells were subsequently collected by centrifugation, washed two times with pre-warmed YEPD supplemented with 100 µg/ml pronase (Roche), and once with pre-warmed YEPD without pronase. Cells were resuspended in pre-warmed YEPD with 15 µg/ml nocodazole (Sigma) and further cultivated at 30 °C. Samples for western blot analysis were taken immediately and every 30 min after release from the α factor arrest.

## Analysis of cell cycle progression after release from a nocodazole arrest

Cells from a saturated overnight culture were diluted to $OD_{600} = 0.2$ and grown at 30 °C for 1.5 h. Nocodazole was added to a final concentration of 15 µg/ml, followed by further incubation at 30 °C for 2 h. Arrest in mitosis was confirmed by microscopy when more than 90% of cells were large-budded. Cells were harvested by centrifugation, washed three time with pre-warmed YEPD, resuspended in pre-warmed YEPD with 10 µg/ml α factor to arrest cells in the next G1 phase, and further cultivated at 30 °C. Samples for western blot analysis were collected every 15 min after release from the nocodazole arrest.

## Viability assay after nocodazole treatment

Cells from a saturated overnight culture were diluted to $OD_{600} = 0.2$ and incubated at 30 °C for 3 h. Nocodazole was added to a final concentration of 15 µg/ml, and cultures were further incubated at 30 °C for 1.5 h. Before and after nocodazole treatment, an aliquot was taken from the cultures and serially diluted to $OD_{600} = 0.0004$. In total, 100 µl of the diluted culture was plated on YEPD plates and incubated at 30 °C for two days. The number of colonies grown on each plate was counted. For each strain, the number of colonies formed in the untreated sample was defined as 100%.

## Live-cell microscopy

For live-cell microscopy to quantify fluorescence signal intensities, cells from a saturated overnight culture were diluted to $OD_{600} = 0.2$

in YEPD medium and grown at 30 °C for four to 5 h. Cells were then immobilized on glass bottom dishes (No 1.5, MatTek corporation) coated with 0.1 mg/ml concanavalin A (Sigma), and the dish was filled with SD doTrp medium.

To analyze chromosome segregation after nocodazole washout, cells were diluted to $OD_{600} = 0.2$ in YEPD medium and grown at 30 °C for 2 h. Cells were treated with 15 µg/ml nocodazole, further incubated at 30 °C for 1.5 h, and then immobilized in multiwell glass bottom dishes (Ibidi) coated with 0.1 mg/ml concanavalin A (Sigma). The Cells were washed three times with SD doTrp to wash out the nocodazole, and the imaging chamber was filled with SD doTrp. Images were acquired at 10 min intervals over a total time of 1.5–2 h. All media were supplemented with 100 µM CuSO$_4$ to induce expression of LacI-GFP which is under control of the Cup1 promoter.

All microscopy was performed with a DeltaVision Elite wide-filed microscope (GE Healthcare Life Sciences) with a Super-Plan Apo 100×/1.4 oil objective, solid state light source (SSCI) and sCMOS camera. Imaging was performed at 30 °C. Z-stacks with a spacing of 0.4 µm covering a total distance of 6 µm were acquired and maximum projections were generated. Alternatively, OAI (optical axis integration) scans over 6 µm were performed. Images were deconvolved using the default settings of the SoftWoRx software (GE Healthcare Life Sciences).

## Quantification of fluorescence microscopy

Fluorescence microscopy signals were quantified using Fiji. Equally sized boxes were drawn around individual kinetochore clusters of metaphase or anaphase cells, and the mean integrated density was measured. In addition, the local background in proximity to the kinetochore cluster was measured and subtracted from the signal.

For each experimental dataset, the mean value measured for metaphase kinetochore clusters of the Kre28$^{WT}$ strain was defined as 1.0 as and all other data was normalized accordingly for better comparison.

To quantify the fraction of anaphase spindles showing kinetochore-proximal Sli15-mNeonGreen foci (Fig. 7F,G), the projected images were blinded and randomized. From these images, the anaphases were classified based on the distribution of Sli15-mNeonGreen along the spindle.

## Statistical analysis

Data was analyzed and visualized using the statistics software GraphPad Prism 10. Mean values +/− standard deviation (SD) are displayed. Information on sample size and applied statistical tests are provided in the figure legends.

## Multiple sequence alignments and secondary structure prediction

Multiple sequence alignments were performed with sequences of Kre28 proteins from different yeast species or with sequences of Kre28 and Zwint-related proteins from various eukaryotic species were used.

Sequences from yeast species were aligned using the MAFFT multiple alignment tool (Katoh et al, 2019). Alignments were visualized in Jalview (Waterhouse et al, 2009) with Clustal coloring and 50% conservation threshold.

Secondary structure predictions of *S. cerevisiae* Kre28 and *H. sapiens* Zwint were generated using the JPred4 server (Drozdetskiy et al, 2015).

## ColabFold structure predictions

The structure of the Spc105$^{490-917}$/Kre28 heterodimer in Fig. 1A was predicted using Colabfold (https://colab.research.google.com/github/sokrypton/ColabFold/blob/main/beta/AlphaFold2_advanced.ipynb) (Mirdita et al, 2022) with the following parameters: msa_method=mmseqs2, pair_mode=unpaired, cov=0, qid=0, max_msa=512:1024, subsample_msa=True, num_relax=0, use_turbo=True, use_ptm=True, rank_by=pLDDT, num_models=5, num_samples=1, num_ensemble=1, max_recycles=3, tol=0, is_training=False, use_templates=False. The figures of the predicted structures were further processed using the software ChimeraX (Pettersen et al, 2004). pLDDT values were colored in ChimeraX. Visualization of the predicted alignment errors was prepared by Colabfold.

## AlphaFold2 molecular modeling

AlphaFold (AF) was used for all molecular modeling in the versions AF2 Multimer 3.2.1 (preprint: Evans et al, 2022) and AF3 (Abramson et al, 2024). A general feature of AF2 Multimer is the higher sensitivity to intra- and intermolecular interactions compared to the original AF2 (Jumper et al, 2021). For the model shown in Fig. 7A,B, the input sequences *S. cerevisiae* Spc105_449-642 (P53148); Kre28_1-208 (Q04431) and Sli15_215-255 (P38283) were used. PAE-scores for this interface were consistently >0.8 and AF2 and AF3 yielded very similar predictions. Segment Sli15 236–248 was predicted with the highest confidence, the Sli15 alpha-helix 224–236 scored slightly lower in pLDDT. The point mutations Sli15 R231E R232E L239E K242E, were designed based on the AF2 prediction using manual inspection as well as a PISA analysis (Krissinel and Henrick, 2007).

## Expression of recombinant proteins in *Sf9* insect cells

KMN and individual KMN subcomplexes were expressed in Sf9 (*Spodoptera frugiperda*) insect cells. Cells were infected with baculoviruses carrying expression constructs for the respective protein complexes. Information about baculoviruses used in this study is listed in the R&T table. For expression of the ten subunit KMN network, cells were co-infected with KT1-5 and KT1-6. Sf9 cells were split to a cell density of $0.8 \times 10^6$ cells/ml and grown at 27 °C with gentle agitation. The next day, cells were infected with the respective baculoviruses and harvested 2–3 days after infection.

## Pull-down assays

For pull-down assays using proteins expressed in Sf9 cells, cells were resuspended in interaction buffer (25 mM HEPES pH 7.4, 200 mM NaCl, 1 mM MgCl$_2$, 2.5% (v/v) glycerol, 0.01% (v/v) Tween-20) supplemented with 1 mM PMSF and Pierce™ Protease Inhibitor (EDTA-free; ThermoFisher Scientific) and lysed by sonication. Lysates were cleared by centrifugation. Anti-Flag M2 agarose beads (Sigma-Aldrich) were preequilibrated with interaction buffer and incubated with the cleared lysate overnight, rotating at 4 °C. Afterward, beads were washed three times for 5 min with interaction buffer, and loaded beads were split into individual pull-down samples.

To test for binding of Ipl1/Sli15 or Ipl1 to KMN or individual subcomplexes, 100 µl of 2 µM Ipl1/Sli15 or Ipl1 was added to ~20 µl of loaded beads. As controls, Ipl1/Sli15 or Ipl1 was also added to unloaded beads or loaded beads were incubated with interaction buffer without any protein. In total, 20 µl of the assembled reactions was taken as input samples for SDS-PAGE analysis. Pull-down samples were subsequently incubated rotating at 4 °C for 6 h. Afterward, beads were collected by centrifugation, the supernatant was removed, and the beads were washed three times with 200 µl interaction buffer. Proteins bound to the beads were eluted by the addition of 50 µl interaction buffer with 0.5 or 1.0 µg/µl 3xFlag peptide, followed by incubation at 4 °C for 1–2 h.

## In vitro phosphorylation and dephosphorylation of Ipl1/Sli15

Recombinant Ipl1/Sli15 was diluted in interaction buffer (25 mM HEPES pH 7.4, 200 mM NaCl, 1 mM MgCl$_2$, 2.5% (v/v) glycerol, 0.01% (v/v) Tween-20) supplemented with 1 mM MnCl$_2$. λ-Protein Phosphatase (NEB) or 500 µM ATP was added, and samples were incubated at 30 °C for 30 min.

## In vitro kinase assay

In vitro kinase reactions were prepared in a total reaction volume of 90 µl. Recombinant KMN, Ipl1/Sli15 or a mixture of KMN and Ipl1/Sli15 were diluted in kinase buffer (20 mM HEPES pH 7.4, 100 mM KCl, 10 mM MgCl$_2$, 10 mM MnCl$_2$, 25 mM β-glycerol phosphate, 1 mM DTT) supplemented with 100 µM cold ATP and 83.25 nM hot $^{32}$P-γ-ATP (Hartmann Analytic). Samples were incubated at 30 °C and 20 µl aliquots were taken after 0, 5, 15, and 30 min. Reactions were stopped by the addition of 6x SDS-PAGE loading buffer, boiled at 95 °C and analyzed by SDS-PAGE. Gels were stained with Coomassie and dried afterward. $^{32}$P autoradiography was detected with a FLA-3000 Imaging Scanner (Fujifilm).

## Purification of recombinant Ipl1/Sli15 from *E. coli*

Ipl1/6xHis-Sli15 (wild type or 4E mutant) was expressed in BL21 *E. coli* cells. 4 L TB medium with 50 µg/ml kanamycin were inoculated with 100 ml of a saturated overnight culture. Cells were incubated at 37 °C, shaking at 180 rpm, until an OD$_{600}$ ≈ 0.7 was reached. Expression of Ipl1/Sli15 was induced by the addition of 0.4 mM IPTG. Cells were further incubated for 3 h and afterward harvested by centrifugation, washed in PBS, and stored at −80 °C until further processing.

Cells were resuspended in 90 ml lysis buffer (50 mM Na$_2$HPO$_4$/NaH$_2$PO$_4$ pH 8.0, 300 mM NaCl, 5% (v/v) glycerol, 20 mM imidazole, 0.5 mM TCEP) supplemented with Pierce™ Protease Inhibitor (EDTA-free; ThermoFisher Scientific) and 1 mM PMSF and lysed by sonication. The lysate was cleared by centrifugation and loaded onto two serially connected HisTrap 5 ml columns (Cytiva) which were connected to a ÄKTA FPLC system (GE Healthcare). Columns were washed with lysis buffer, and bound proteins were eluted with a linear imidazole gradient (20–500 mM). Elution fractions were collected and analyzed by SDS-PAGE. Fractions of comparable quality were pooled, concentrated to ~500 µl, and subjected to size exclusion chromatography using Superdex 200 10/300 GL column (GE Healthcare; 20 mM HEPES pH 7.4, 250 mM NaCl, 1 mM MgCl$_2$, 5% (v/v) glycerol, 0.5 mM

TCEP). Elution fractions were analyzed by SDS-PAGE and comparable samples were pooled, aliquoted, snap-frozen in liquid $N_2$, and stored at $-80\,°C$.

### In vivo biotinylation and affinity purification with TurboID

Cell extracts from the respective yeast strains were prepared from log-phase cultures. Cells were lysed by bead-beating (BioSpec) in Lysis buffer (20 mM HEPES pH 7.4, 200 mM NaCl, 1 mM $MgCl_2$, 0.5 mM TCEP, 2.5% glycerol. After centrifugation, 500 µl cleared lysate was incubated with 50 µl preequilibrated streptavidin-sepharose beads (Cytiva) with rotation at 4 °C overnight. Beads were washed twice with wash buffer (Lysis buffer + 0.05% (v/v) TX-100), twice with wash buffer + 2% (w/v) SDS, twice with wash buffer + 0.1% (w/v) SDS and finally twice with wash buffer. Proteins were eluted off the beads with SDS-PAGE loading buffer and analyzed by western blotting.

### Pt/C low-angle rotary shadowing electron microscopy

The peak elution fractions from gel filtrations of recombinant KMN complexes were further prepared for rotary shadowing EM analysis by diluting the complexes to ~70 µg/ml in spraying buffer, containing 100 mM ammonium acetate and 30% (v/v) glycerol, pH adjusted to 7.4. After dilution, the samples were sprayed onto freshly cleaved mica chips and immediately transferred into a Bal-Tec MED020 high vacuum evaporator equipped with electron guns. After drying in the vacuum, the rotating samples were coated with 0.5 nm platinum at an elevation angle between 5° and 6°, followed by 7 nm carbon at 90°. The produced replicas were floated off from the mica chips and picked up on 400 mesh Cu/Pd grids. The grids were inspected in an FEI Morgagni 268D TEM operated at 80 kV. Electron micrographs were acquired using an 11 megapixel Morada CCD camera from Olympus-SIS. Fiji was used for image analysis.

### Chemical cross-linking and mass spectrometry

In total, 50–60 µg of recombinant protein complexes were cross-linked with 1:1 isotopically-labeled ($d_0/d_{12}$) BS3 (Bis[sulfosuccinimidyl] suberate, Creative molecules) in a Thermomixer at 35 °C for 30 min at 1200 rpm. The reaction was quenched with 100 mM ammonium bicarbonate for 5 min at room temperature and supplemented with 8 M urea to a final concentration of 6 M. After reduction and alkylation cross-linked proteins were digested with Lys-C (1:50 w/w, Wako) for 3 h, diluted with 50 mM ammonium bicarbonate to 1 M urea and digested with trypsin (1:50 w/w, Promega) overnight. Cross-linked peptides were purified by reversed-phase chromatography using C18 cartridges (Sep-Pak, Waters). Cross-linked peptides were enriched by peptide size exclusion chromatography and analyzed by tandem mass spectrometry (Orbitrap Elite, Thermo Scientific). Fragment ion spectra were searched, and crosslinks were identified by the dedicated software program xQuest (Walzthoeni et al, 2012).

## Data availability

The mass spectrometry proteomics chemical cross-linking data have been deposited to the ProteomeXchange Consortium via the PRIDE (Perez-Riverol et al, 2022) partner repository with the dataset identifier PXD059609. Link: https://www.ebi.ac.uk/pride.

The source data of this paper are collected in the following database record: biostudies:S-SCDT-10_1038-S44318-025-00437-w.

## Peer review information

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

## Acknowledgements

This work was funded by the Deutsche Forschungsgemeinschaft (DFG, German Research Foundation)– SFB1430–Project-ID 424228829. Microscopy experiments were carried out with support from the Imaging Center Campus Essen Core Facility (ICCE). The DeltaVision Elite high-resolution microscope was obtained through Deutsche Forschungsgemeinschaft funding (Major Research Instrumentation Program as per Art. 91b GG, INST 20876/275-1). The laboratory of FH was supported by the European Research Council (ERC)-StG (638218) and by the GFF NÖ (Endowed Professorship, Province of Lower Austria).

## Author contributions

**Alexander Dudziak**: Conceptualization; Formal analysis; Investigation; Visualization; Writing—original draft; Writing—review and editing. **Richard Pleuger**: Conceptualization; Formal analysis; Investigation; Visualization; Writing—review and editing. **Jasmin Schmidt**: Investigation. **Frederik Hamm**: Investigation. **Sharvari Tendulkar**: Investigation. **Karolin Jänen**: Investigation. **Ingrid R Vetter**: Formal analysis; Validation; Investigation; Visualization. **Sylvia Singh**: Investigation. **Josef Fischböck**: Investigation. **Franz Herzog**: Formal analysis; Supervision; Funding acquisition; Investigation. **Stefan Westermann**: Conceptualization; Formal analysis; Supervision; Funding acquisition; Investigation; Writing—original draft; Project administration.

Source data underlying figure panels in this paper may have individual authorship assigned. Where available, figure panel/source data authorship is listed in the following database record: biostudies:S-SCDT-10_1038-S44318-025-00437-w.

## Funding

## Disclosure and competing interests statement

The authors declare no competing interests.

# Expanded View Figures

**Figure EV1.   Refers to Fig. 1: Detailed mutagenesis analysis of the Zwint helix.**

(**A**) Multiple sequence alignment of Kre28 sequences from different yeast species. Conserved residues are colored according to the Clustal color scheme. Positions of smaller Zwint helix deletions and alanine substitutions of conserved residues are marked. (**B**) Western blot analysis of different Kre28/Zwint deletion or point mutants. Extracts of log-phase yeast cells were analyzed, Pgk1 blot serves as a loading control. (**C, D**) Serial dilution assays of haploid yeast strains with different Kre28 Zwint helix deletions or point mutations of conserved residues. Strains with the indicated genotypes were serially diluted and spotted on YEPD or YEPD + benomyl plates and incubated at the indicated temperatures.

▶

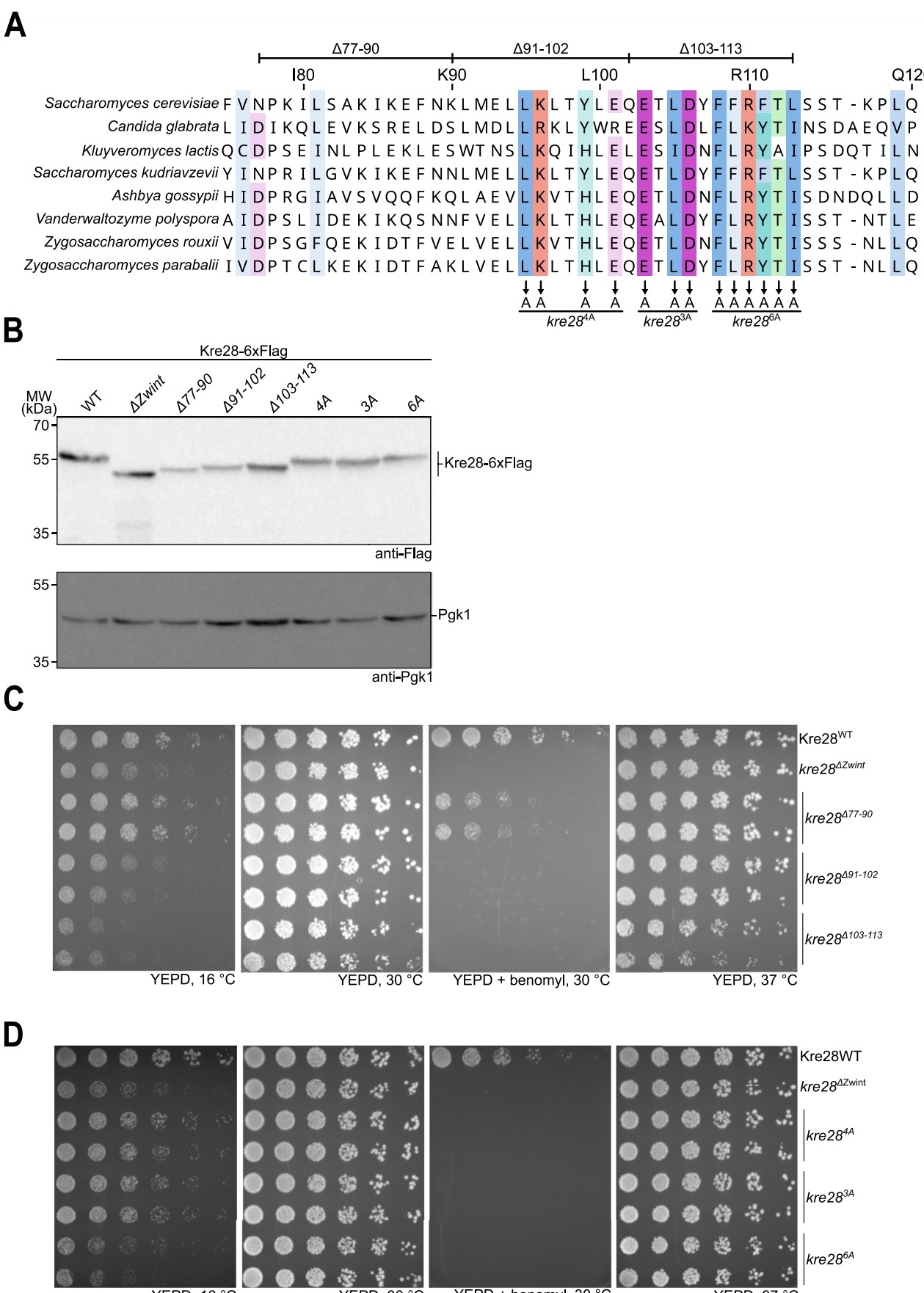

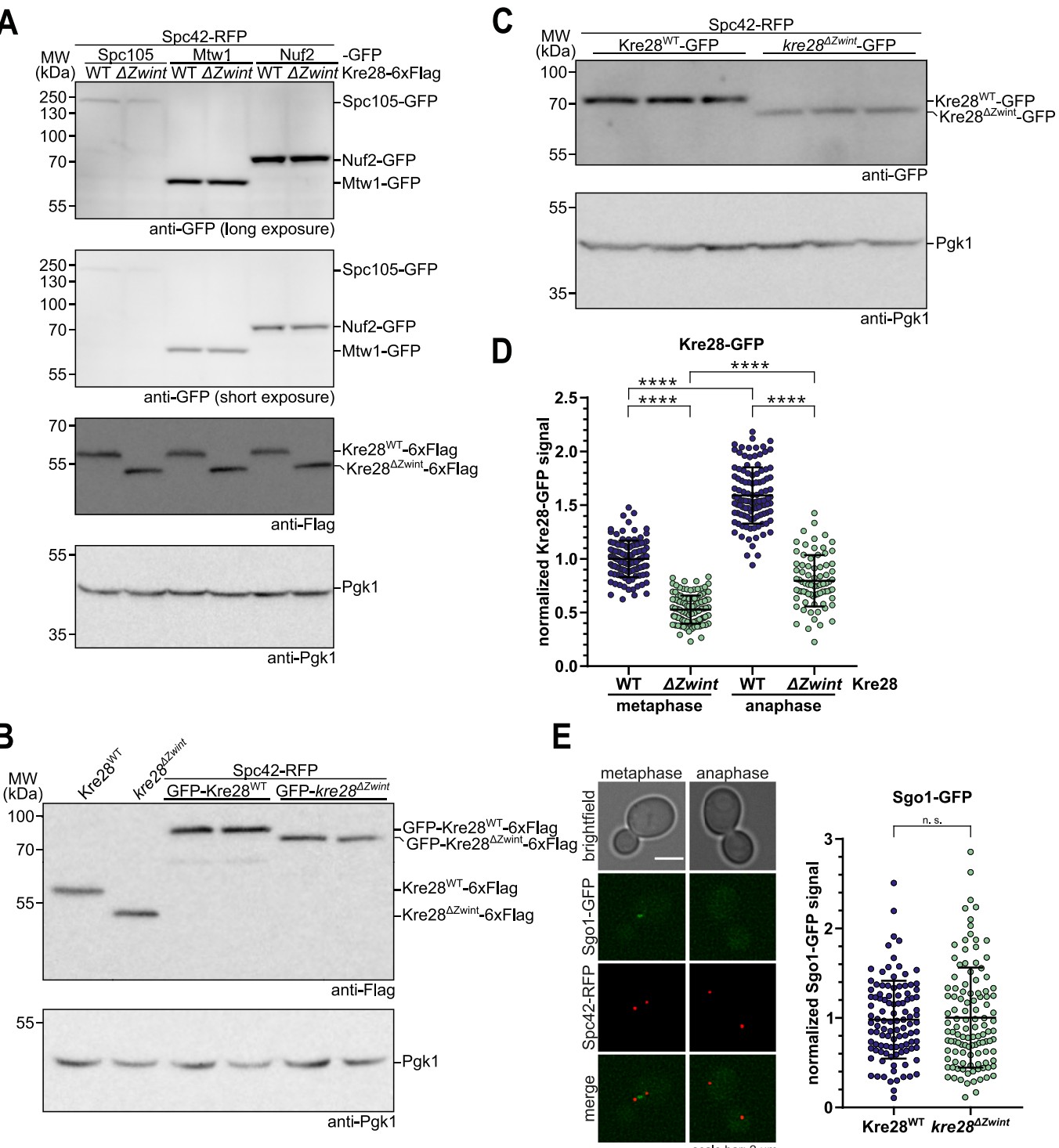

◀ **Figure EV2. Refers to Fig. 3: Western blot analysis of GFP-tagged KMN subunits and live-cell microscopy of Sgo1-GFP.**

(A) Western blot analysis to compare the expression of different GFP-tagged KMN subunits. in Kre28 wild-type and mutant cells. Pgk1 served as loading control. (B) Western blot analysis of expression level of N-terminal GFP-tagged Kre28 in wild-type and mutant form. (C) Western blot analysis of C-terminal GFP-tagged Kre28 in wild-type and mutant form. (D) Quantification of Kre28-GFP fluorescence signals at metaphase and anaphase kinetochores. Strains with either Kre28$^{WT}$-GFP or $kre28^{\Delta Zwint}$-GFP were analyzed by live-cell microscopy. The C-terminus of Spc42 was fused to RFP to label the position of the mitotic spindle. The mean of metaphase signal intensities in the wild-type strain was defined as 1 and all other data was normalized to this value. Means $+/-$ standard deviation are plotted. $P$ values were calculated with a Kruskal–Wallis test, and displayed as follows: ****$P < 0.0001$. Exact $P$ values for (E): in all cases ****$<0.0001$. $n > 100$ kinetochore clusters were analyzed for each condition. Representative data from two independent experiments is shown. (E) Live-cell microscopy for analysis of Sgo1-GFP kinetochore localization in a Kre28$^{WT}$ or $kre28^{\Delta Zwint}$ strain background. Representative images of a metaphase and an anaphase cell are shown on the left and quantification of Sgo1-GFP signals in cells with short mitotic spindles is shown on the right. Data was normalized to the mean value measured for the Kre28$^{WT}$ strain. Mean values $+/-$ standard deviation are plotted. A Mann–Whitney test was used to test for statistical significance.

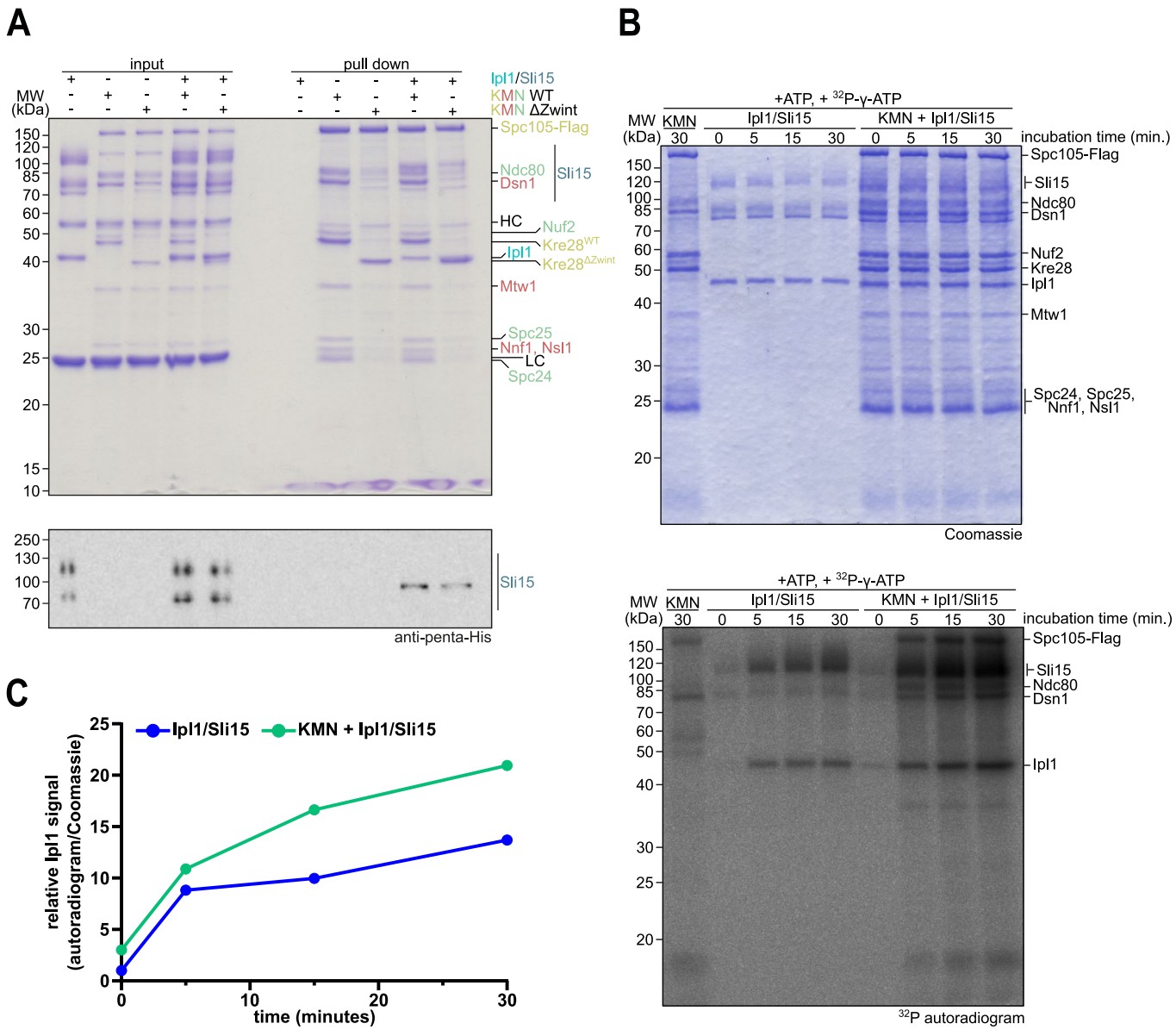

**Figure EV3.  Refers to Fig. 5: Pull-down experiment and in vitro kinase reaction with Ipl1/Sli15 and KMN.**

(A) Pull-down assays to analyze binding of Ipl1/Sli15 to KMN. KMN with either Kre28[WT] or Kre28[ΔZwint] was purified from Sf9 cells and immobilized on M2 anti-Flag beads. Loaded beads were incubated with buffer or recombinant Ipl1/Sli15, afterwards proteins bound to the beads were eluted with 3xFlag peptide. Input and pull-down samples were analyzed by SDS-PAGE and Coomassie staining or western blot to detect 6xHis-tagged Sli15. (B) In vitro kinase reaction with Ipl1/Sli15 or Ipl1/Sli15 in the presence of recombinant KMN analyzed over time. Coomassie-stained gel at the top, corresponding autoradiograph at the bottom. For control a KMN only sample taken 30 min after incubation with radioactive ATP is shown in the first lane. (C) Quantification phosphorylated Ipl1 signal over time alone (blue curve) or in the presence of KMN (green curve).

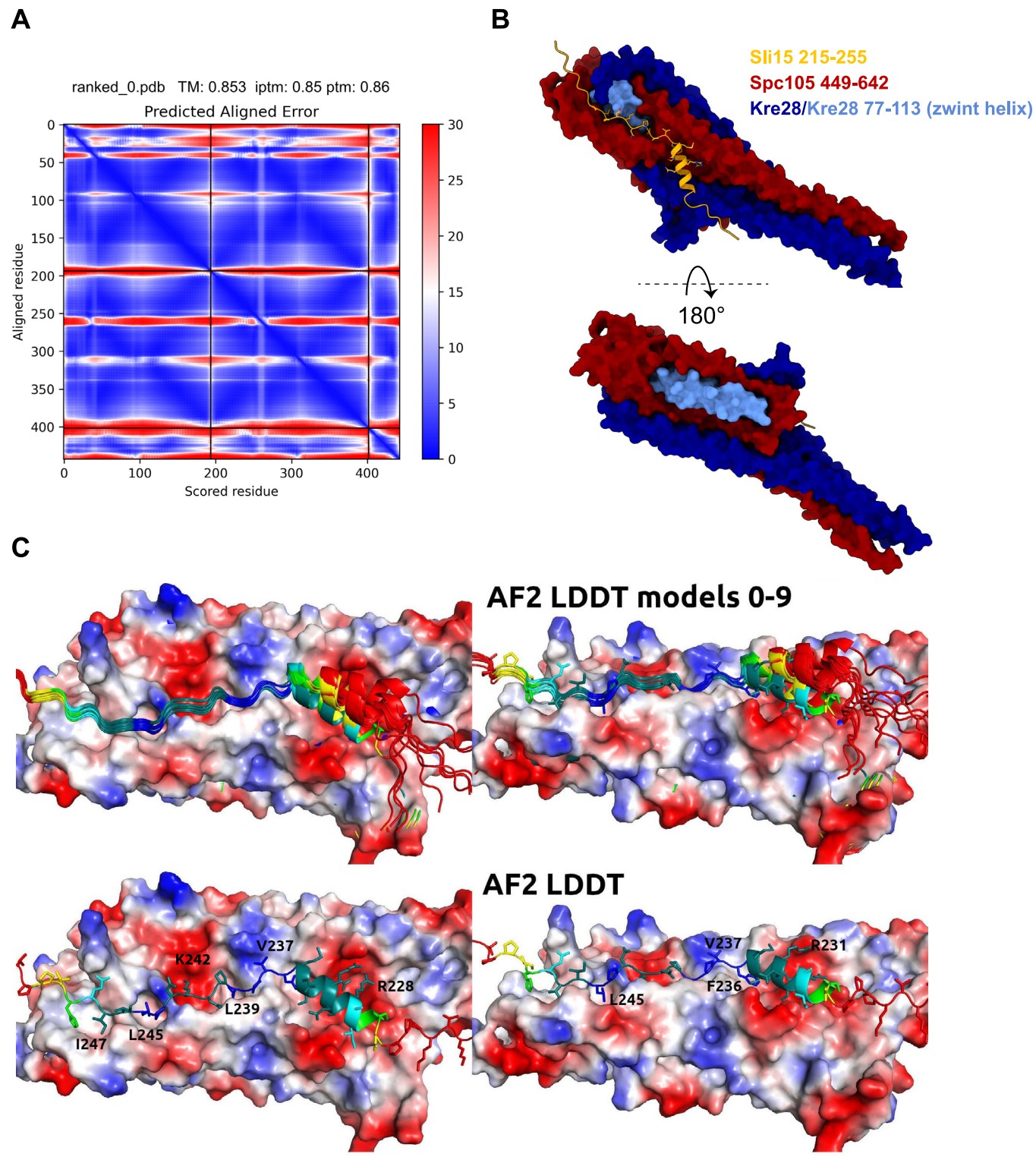

◀ **Figure EV4.   Refers to Fig. 7.**

(A) PAE plot of the top-ranked model of the Spc105 (449–642)-Kre28 (1–208)- Sli15 (215–255) complex. Fragment boundaries are indicated by black lines, blue are favorable scores, red are unfavorable. (B) Position of the Kre28 Zwint helix (light blue) in the top-ranked model. Note that only a part of the Zwint helix is involved in the binding interface with Sli15, while the majority of the helix is on the opposite side of the Spc105/Kre28 bundle. (C) Top row: Superposition of the ten top-ranked models of the Sli15 peptide on Spc105/Kre28. pLDDT scores for the Sli15 peptides are indicated with blue and green showing favorable, and red unfavorable scores. The Spc105/Kre28 surface is colored by electrostatic potential. Bottom: Details of the top-ranked model, with key side chains of the Sli15 peptide indicated as sticks. Coloring by pLDDT scores as above.

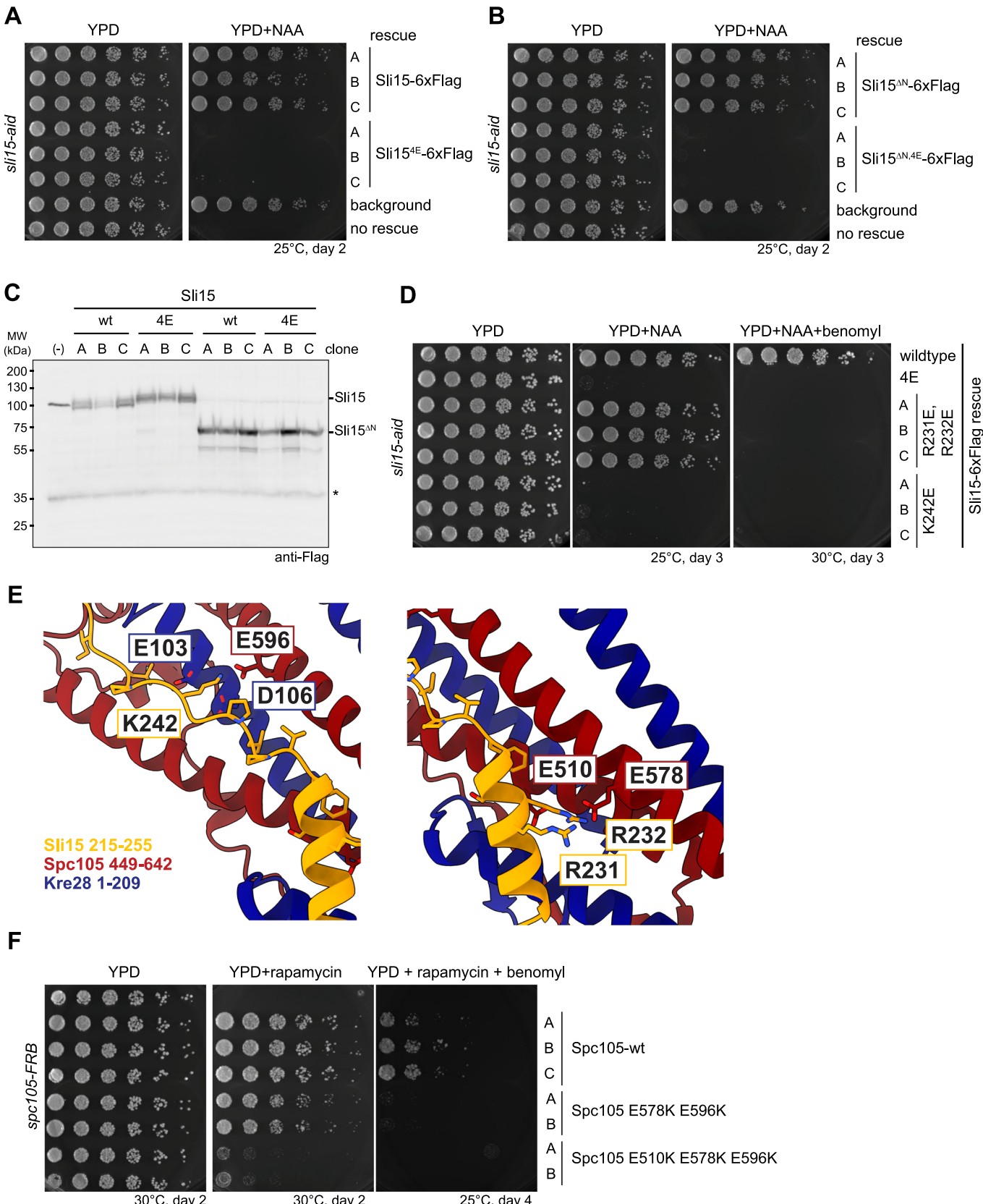

◀  **Figure EV5.   Refers to Fig. 7.**

(A) Serial dilution assay testing full-length Sli15 wild-type and -4E rescue constructs in a *sli15-aid* system. Three clones for wild type and mutant are tested. Background indicates a strain lacking *sli15-aid*, no rescue indicates a *sli15-aid* strain without a Sli15 rescue construct. (B) Serial dilution assay testing Sli15-ΔN wild-type and -4E rescue constructs in a *sli15-aid* system. Three clones for each version are tested. Background indicates a strain lacking sli15-aid, no rescue indicates a sli15-aid strain without a Sli15 rescue construct. (C) Western Blot analysis of Sli15-Flag rescue constructs tested in A and B. Asterisk denotes unspecific band. (D) Serial dilution assay testing the indicated Sli15 rescue constructs in a *sli15-aid* system. Three clones for each version (Sli15 R231E R232E, or Sli15 K242E) are tested. (E) Left side: Details of the predicted binding site surrounding Sli15 K242 (orange) with interacting residues in Kre28 (dark blue) or Spc105 (dark red) highlighted. Right side: Details of the predicted binding site surrounding Sli15 residues R231 and R232. (F) Serial dilution assay testing the effects of Spc105 mutations predicted to affect the Sli15-binding site. Spc105 rescue constructs are tested in an Spc105-FRB anchor-away system in which upon rapamycin addition the endogenous Spc105 protein is removed from the nucleus. Top row: control lacking a rescue construct. Three clones for wild-type rescue constructs, and two clones of each mutant are tested.

