## [Peer Review File · The EMBO Journal]

Spc105/Kre28 promotes mitotic error correction by outer kinetochore recruitment of Ipl1/Sli15

Alexander Dudziak, Richard Pleuger, Jasmin Schmidt, Frederik Hamm, Sharvari Tendulkar, Karolin Jänen, Ingrid Vetter, Sylvia Singh, Josef Fischboeck, Franz Herzog, and Stefan Westermann

Corresponding author(s): Stefan Westermann (stefan.westermann@uni-due.de)

Review Timeline:

Submission Date:	4th Oct 24
Editorial Decision:	5th Nov 24
Revision Received:	17th Feb 25
Editorial Decision:	27th Mar 25
Revision Received:	31st Mar 25
Accepted:	2nd Apr 25

Editor: Hartmut Vodermaier

Transaction Report:

Prof. Stefan Westermann
University of Duisburg-Essen
Department of Molecular Genetics, Faculty of Biology
Centre of Medical Biotechnology
Essen 45117
Germany

5th Nov 2024

Re: EMBOJ-2024-119215
SpC105-Kre28 promotes mitotic error correction by kinetochore recruitment of lpl1/Aurora B kinase

Dear Stefan,

Thank you for submitting your manuscript on SpC105-Kre28 as a new Aurora B kinase receptor for our consideration. I sent it to three expert referees, who have now returned the reports that are copied below. As you will see, the referees consider your findings interesting and potentially important, but also raise several substantive criticisms that would have to be clarified prior to publication. In particular, all reviewers share concerns regarding the strength of certain biochemical analyses, and the conclusiveness of the AlphaFold models at the current stage.

Should you be able to adequately address the referees' comments, we would be happy to pursue a revised manuscript further for EMBO Journal publication. Please be reminded, however, that our single-major-revision-round policy makes it important to diligently respond to each referee point at the time of resubmission. I would therefore encourage you to contact us with a revision plan, preliminary point-by-point response, or any other questions you may have in this regard already during the early stages of your revision work, in order to clarify if and how key issues raised in the reports may be solved. We would also be open to extension of the regular three-months revision period if needed; our 'scooping protection' (meaning that competing work appearing elsewhere in the meantime will not affect our considerations of your study) would of course remain valid also throughout such an extension.

Further information on preparing, formatting and uploading a revised manuscript can be found below and in our Guide to Authors. Thank you again for the opportunity to consider this work for The EMBO Journal, and I look forward to hearing from you in due time.

With kind regards,

Hartmut

3) Revised manuscript text (including main tables, and figure legends for main and EV figures) has to be submitted as editable

text file (e.g., .docx format). We encourage highlighting of changes (e.g., via text color) for the referees' reference.

4) Each main and each Expanded View (EV) figure should be uploaded as individual production-quality files (preferably in .eps, .tif, .jpg formats). For suggestions on figure preparation/layout, please refer to our Figure Preparation Guidelines:

8) Please note that supplementary information at EMBO Press has been superseded by the 'Expanded View' for inclusion of additional figures, tables, movies or datasets; with up to five EV Figures being typeset and directly accessible in the HTML version of the article. For details and guidance, please refer to:

embopress.org/page/journal/14602075/authorguide#expandedview

9) To facilitate reproducibility and cross-laboratory adoption of methodologies, please structure the Materials & Methods section as outlined in our guide to authors, including a completed Reagents and Tools Table that can be downloaded from our author guidelines as well (<https://www.embopress.org/page/journal/14602075/authorguide#structuredmethods>).

10) Digital image enhancement is acceptable practice, as long as it accurately represents the original data and conforms to community standards. If a figure has been subjected to significant electronic manipulation, this must be clearly noted in the figure legend and/or the 'Materials and Methods' section. The editors reserve the right to request original versions of figures and the original images that were used to assemble the figure. Finally, we generally encourage uploading of numerical as well as gel/blot image source data; for details see: embopress.org/page/journal/14602075/authorguide#sourcedata

At EMBO Press, we ask authors to provide source data for the main manuscript figures. Our source data coordinator will contact you to discuss which figure panels we would need source data for and will also provide you with helpful tips on how to upload and organize the files.

Further information is available in our Guide For Authors:

In the interest of ensuring the conceptual advance provided by the work, we recommend submitting a revision within 3 months (3rd Feb 2025). Please discuss the revision progress ahead of this time with the editor if you require more time to complete the revisions. Use the link below to submit your revision:

Link Not Available

Referee #1:

Dudziak et al. present extensive data documenting the presence of a new binding site for the Aurora B/Ipl1 kinase in the budding yeast kinetochore. The Aurora B kinase, hereon referred to as Ipl1, is the master regulator of the error correction pathway, which dissolves erroneous or syntelic chromosome attachments to the spindle. It has long been theorized that the error correction pathway is mediated by Ipl1 recruitment sites at the centromere and the centromere-proximal regions of the kinetochore. The new data strongly suggest that the Spc105-Kre28 complex located in the "outer" kinetochore harbors an additional Ipl1 binding site that makes significant contributions error correction in budding yeast. Thus, these findings will influence future models of the error correction mechanism and should generate strong interest.

The authors are to be commended for presenting extensive and multi-pronged data to discover and characterize the interaction between the Spc105-Kre28 and Ipl1-Sli15 complexes. They offer conclusive evidence showing that: (a) the coiled-coil section of the Spc105-Kre28 complex is important for accurate chromosome segregation in a manner consistent with establishing proper Ipl1 activity in the kinetochore, (b) the Spc105-Kre28 can interact with the Ipl1-Sli15 complex in vitro, and this interaction is sensitive to Ipl1 autophosphorylation, (c) artificial recruitment of Ipl1 in a similar site as Spc105-Kre28 restores cell viability in

Kre28 mutants. Overall, the experiments are well-designed and executed but, the picture that emerges contains some significant gaps discussed below. Nevertheless, this work achieves a significant advance in our understanding of the error correction mechanism; it will guide future efforts to fully defining the mechanism of error correction. Therefore, I support its publication after the authors have addressed the following issues.

Main comments:

1. What is the exact nature of the Ipl1/Sli15 binding site in Spc105-Kre28? The evidence for the new Ipl1-Sli15 binding site in the Spc105-Kre28 complex is not definitive. While the authors perform extensive and careful pull-down experiments with recombinant proteins, I am unable to rationalize the following incongruity. The conserved "Zwint-1" helix in Kre28 is clearly important for accurate chromosome segregation, however, the AlphaFold predictions (Fig. 1) show that it is located distal to the interaction site for Ipl1 predicted in Fig. 2. Interestingly, deletion of the Zwint-1 helix does not appear to have any effect in the in vitro pull-down assay shown in Figure 4c. Furthermore, the chemical crosslinking experiment also did not yield any hits with this section of Zwint-1. The authors should offer the best possible explanation for why this might be so. For example, does the deletion of this helix simply disrupt the Ipl1-binding interface? Is the reduced amount of Spc105 in this mutant result in reduced Ipl1 recruitment? I suggest that the authors fully discuss the implications of their findings to help the reader in achieving a good understanding of the work.

2. The synthetic interactions of the Ipl1 N-terminal mutants are intriguing and consistent with the authors' model. However, it remains possible that other Ipl1 interactions mediated by the residues are responsible for the phenotype. The authors should test whether artificial recruitment of Ipl1 used earlier can rescue viability of the *ipl1-3A ctf19-RWDdelete* mutant.

3. In the same vein, it is important to perform one additional experiment because AF2 predicts the Ipl1-Spc105 interaction with very low confidence. The authors should disrupt the binding interface in Spc105 by mutating residues predicted by the AF2 models to interact with the Ipl1 N-terminus. Genetic interactions of this mutant with Sgo1-AID or *ctf10-RWDdelete* mutants will significantly strengthen the authors' case.

4. A predicted structure of Ipl1 and Spc105-Kre28 that highlights the interaction surface on Spc105-Kre28 as well as the location of the Zwint-1 helix in Kre28 would be very useful to help the readers understand the nature of the putative interaction site. Surface color depicting a conservation score would also be quite useful to understand whether the yeast results can be generalized easily to other eukaryotes.

Minor comments:

1. Is it possible that the two appendages in Fig. 4B correspond to the MIND and Spc105-Kre28 coiled-coil domains respectively?
2. Indicate the coiled coil domains of Spc105 and the Zwint-1 helix in Kre28 in Figure 5.
3. Label the bands corresponding to Ipl1 and Sli15 in Figure 7.
4. The PAE plots in the supplementary Figure 7 are too small to study especially for the small section of Ipl1. The PAE plot for the best model should be presented by itself and made legible.
5. It would be useful to use the colony color assay to evaluate the contributions of the Zwint-1 helix to accurate chromosome segregation under normal conditions. This is optional because it adds context that is not necessary to support the main advance of this work.

Referee #2:

The manuscript "The Spc105Knl1/Kre28Zwint complex promotes mitotic error correction by outer kinetochore recruitment of the Ipl1Aurora B kinase" by Dudziak et al explores an important topic for the kinetochore and mitosis fields. Determining the outer kinetochore receptor of Ipl1 is a significant topic and would provide needed insight into mechanisms of kinetochore-microtubule error-correction in particular. In their manuscript, Dudziak and colleagues begin by characterizing a kre28 mutant in a conserved alpha helical region proximal to the Spc105-binding alpha helical domain. A deletion of this "zwint helix" renders cells sensitive to cold temperatures and the microtubule poison benomyl and has phenotypes consistent with defective mitotic error correction, including synthetic viability defects with *ipl1-1* and chromosome segregation defects. They also show that this *kre28Δzwint* mutant has a proficient SAC but results in reduced Kre28 and Spc105 recruitment to kinetochores. They further show that Ipl1/Sli15 can directly bind to Spc105/Kre28 in vitro to localize the kinase to the KMN network. Crosslinking mass spectrometry confirmed an interaction between Ipl1, Sli15, and Spc105/Kre28 in vitro. Significantly, they showed that a Mtw1-Sli15 fusion was able to rescue the phenotypes of *kre28Δzwint*, in support of the notion that Spc105/Kre28 recruit Ipl1 to the kinetochore. They further propose that the Ipl1 N-terminus comprises the Spc105/Kre28 binding module and present *ipl1* mutants with biochemical and cell phenotypes that are consistent with this idea. Overall, the authors propose a reasonable model that is supported by their data. They are very careful in their writing and do not overclaim what their data show. The Mtw1-Sli15 fusion rescue experiments are superb and are the highlight of the paper, and by contrast many of the biochemical assays are weaker and less clear. However, we recognize the sheer number of biochemical assays shown, and the work this represents, and note that the phenotypes while weak are at least mostly consistent with their model. The data shown in in Figure 7 is weak, in particular the AlphaFold 2 prediction shown in more detail in the supplement, but their mutational analysis is consistent with the conclusions

they draw.

Major points:

1. The structural prediction in Figure 7B is not of high confidence based on the pLDDT metrics and aligned error provided in the supplement. The multiple predicted binding modes shown also do not provide confidence that this structure prediction is correct. I suggest the authors try this prediction again with a larger region of Ipl1 to exclude the possibility that another region contributes to Spc105/Kre28 association. In the absence of this, perhaps a prediction carried out using AlphaFold 3 might also result in a higher confidence model. Alternatively, demonstrating that mutations on the predicted Spc105/Kre28 side of the interface produce a similar phenotype to the *ipl1* mutants would strengthen the authors' claim.
2. The mutational analysis in Figure 7E-G does not strictly support the notion that the Ipl1 N-terminal region binds to the outer kinetochore. It is possible that simply combining multiple slightly deleterious mutations in different pathways results in more severe phenotypes. A rescue of an IPL1 mutant via tethering to MTW1 as in Figure 6C would provide better support to their conclusions.

Minor Points

1. The order of presentation of the data leads a reader to think early on that *kre28Δzwint* may specifically ablate Ipl1 binding at the kinetochore. This leads to confusion in Figure 4 when it becomes apparent that *Kre28Δzwint* can bind Ipl1 similar to WT in vitro. While the authors are careful in stating the loss of Ipl1 in *kre28Δzwint* is due to reduced Mtw1-Spc105/Kre28 association, I think a reordering of the data would better convey what they are trying to say. I propose a reordering of the data to place the figures as written in a new order (Figure 1, 2, 6, 4, 5, 7). Moving the Figure 6 data closer to Figure 2 will help better convey the *kre28Δzwint* phenotypes are due to loss of Ipl1, and that this loss is likely due to reduced association of Spc105/Kre28 within KMN. The authors should clarify that they aim to identify a more specific mutant that effectively delocalizes Ipl1 from Spc105/Kre28 to test their prediction more robustly. Finding such a mutant - one that maintains KMN assembly while excluding Ipl1 - should be noted as a priority, even if not essential for the current study.
2. The crosslinking mass spectrometry data in Figure 5 do not show any interaction between the Ipl1 N-terminal region and Spc105/Kre28. In line 453 the authors state the crosslinking data supports the notion that the Ipl1 N-terminal region contacts Spc105/Kre28. They should remove this language and comment on why they think such crosslinks do not exist if they propose this region to mediate Ipl1-KMN interaction. In addition, this crosslinking figure seems disconnected from the rest of the manuscript and could also be reordered to be in or near Figure 1 in support of the notion that Spc105/Kre28 recruit Ipl1.
3. While very exhaustive, some biochemical analyses show weaker than expected phenotypes especially given the robust cellular response to mutational analysis. In particular, Figures 3C-D appear relatively weak compared to the strong loss of Spc105/Kre28 at the kinetochore observed in cells. While the quantification in 3D helps, the gel itself is not very compelling (for example, the Mtw1 band shows little noticeable difference). Minimally, the authors should comment on why more pronounced differences were not observed. Additionally, there is disagreement between Figure 4C and the accompanying supplement with regards to Ipl1 binding. We propose the authors move 4C to supplement and instead rely on Figure 4D and 4F to conclude Spc105/Kre28 recruit Ipl1 to KMN in vitro.
4. If *Kre28Δzwint* disrupts binding to Mtw1, but still binds Ipl1, you might expect in vivo biotinylation with Spc105-TurboID to show labeling of Sli15 but not an Mtw1-FLAG construct. Completing this experiment would lend confidence to weaker in vitro results.
5. Can a direct Ipl1 in vivo localization assay be performed for any of the mutants tested? Perhaps a Co-IP or fluorescence microscopy.
6. In Figure 2E is it possible to tell whether mis-segregation had any preference for the mother or daughter cell? Providing this information would also provide confidence that *kre28Δzwint* phenotypes are consistent with an *ipl1* mutant phenotype.
7. In Figure 6C Sli15 point mutants fusions should be on the same plate as the Sli15-WT fusions.
8. Very minor, can the authors mention why their *sgo1-AID* strain is using pGAL-TIR?
9. Can the authors label the Zwint helix on Figure 5 and Figure 7? Also, can the authors label residue numbers on Ipl1 in Figure 5?
10. Line 108: typo "INCEP" should be "INCENP"

Referee #3:

Dudziak et al describe a novel allele of yeast Kre28 that highlights a role for Spc105/Kre28 in error correction by facilitating Ipl1 recruitment to kinetochores. The manuscript is very well written, suitable for a broad audience. The work is of high importance since, despite a wealth of studies, how exactly Aurora B regulates error correction remains incompletely understood. Recent studies have highlighted multiple pools of the CPC besides the classical inner centromere and the presence of an (outer)-kinetochore pool has been suggested. However, how the CPC interacts with outer kinetochore has remained elusive. Identifying Spc105/Kre28 as a possible receptor opens a new avenue of research to investigate how the CPC regulates kt-mt attachments, both in yeast as in mammals. However, in its current form the manuscript also contains some serious shortcomings undermining the molecular understanding of the identified *Kre28Δzwint* allele.

The authors model the Spc105-Kre28 complex using alphafold2. Using S.c. Spc105490-917 and FL Kre28 I was able to reproduce the structure they present in Fig 1A. Importantly, this structure scores poorly with pTM and ipTM scores of 0.44. Per

alphafold: "A pTM score above 0.5 means the overall predicted fold for the complex might be similar to the true structure." "ipTM measures the accuracy of the predicted relative positions of the subunits within the complex. Values higher than 0.8 represent confident high-quality predictions, while values below 0.6 suggest likely a failed prediction.". Moreover, closer inspection of the model reveals multiple topological intertwinements that effectively do not appear in experimentally determined structures. This is a known problem for AlphaFold predictions of complexes associated with poor model quality, see for example refs (Dabrowski-Tumanski & Stasiak, 2023; Hou et al, 2023; Lensink et al, 2023). Taken together, the presented structural model is very likely incorrect and inspires little confidence as a molecular basis for interpreting interactions.

The authors proceed to focus on a helical fragment in Kre28 they name the Zwint helix (residues 77-113). Importantly, the Zwint helix highlights the above mentioned problems with topological intertwinement as residues (114-126), C-terminal of the Zwint helix thread through a looped region formed by Spc105 residues 490-640.

Regardless, deletion of the Zwint helix in combination with the RWD domain in Kre28 impairs viability and Kre28 Δ Zwint confers benomyl hypersensitivity and cold sensitivity to cells, indicating a possible role in error correction. Importantly, the authors demonstrate an interaction between the KMN network and the error correction kinase Ipl1 in complex with Sli15.

The authors demonstrate that the recombinant Ipl1/Sli15 complex is (auto)phosphorylated when purified from E. coli. They then demonstrate this autophosphorylated complex binds to the KMN network, specifically to the Spc105/Kre28 subcomplex (Fig 4C,D). Thus, clearly autophosphorylated Sli15-Ipl1 can bind to the KMN network. Now, the recombinant complexes are incubated with ATP or lambda phosphatase and after 30 min at 30 degr. 100 ul (per methods section 'Pull down assays') Ipl1/Sli15 mix is added to 20 ul preloaded (with KMN network) beads. These beads are now incubated for another 6 hours at 4 degr prior to wash and elution. It is unclear how the authors now conclude that autophosphorylation of Ipl1/Sli15 is the sole reason for diminished binding in the ATP samples. During incubations it is highly likely that the KMN network is phosphorylated in the ATP samples. I don't think the authors can rule out that KMN phosphorylation contributes to the diminished binding of Ipl1/Sli15. To prove it is specifically autophosphorylation of Ipl1/Sli15 would require this complex to be further purified after ATP/Lambda phosphatase incubation such that the ATP/Lambda phosphatase only 'sees' Ipl1/Sli15. It is also unclear what the phosphorylation status of the KMN prep from insect cells is? Importantly, phosphorylation of KMN and autophosphorylation of Ipl1/Sli15 need not be mutually exclusive as a mode of regulating an Ipl1/Sli15 pool at the kinetochore.

The authors reason that Spc105/Kre28 is important for Sli15/Ipl1 recruitment to kinetochores. Importantly, Kre28 Δ Zwift results in reduced levels of Spc105/Kre28 and hence Sli15/Ipl1 at kinetochores. However, constitutive recruitment of Sli15/Ipl1 to kinetochores via fusion to Mtw1 only partially rescues missegregation rates in cells expressing the Kre28 Δ Zwift allele. The authors should really emphasize the partial nature of the rescue here. Clearly, the fusion is not nearly enough with close to 40% of cells still missegregating chromosome XII. Importantly, the authors demonstrate Sli15/Ipl1 recruitment to kinetochores is likely phosphoregulated, thus constitutive recruitment may not be expected to fully rescue the Kre28 Δ Zwift allele as it does not permit a dynamically regulated pool of Sli15/Ipl1. This needs to be discussed more clearly. Furthermore, how do Sli15/Ipl1 levels change at kinetochores between WT, Kre28 Δ Zwift and Mtw1-Sli15? Importantly, can the authors rule out that placing 'extra' Ipl1 activity results in a specific rescue by increasing MT turnover at kinetochores? To drive home the point that loss of Spc105/Kre28 in Kre28 Δ Zwift cells results in loss of a kinetochore pool of Sli15/Ipl1 the authors should demonstrate a rescue by reconstituting kinetochore levels of Spc105/Kre28, for example by tethering Kre28 or Kre28 Δ Zwift to Mtw1.

The authors further focus on a potential role of the Ipl1 Nterminus in the interaction with Spc105-Kre28, based in part on previous data that suggest a region around residue 45-55 in binding to Ndc80. Again, the authors use AlphaFold modeling to study how Ipl1 1-30 binds to Spc105-Kre28 but now also in the context of the kKMN network. Importantly, now a larger fragment of Spc105 is chosen, residues 421-917. This complex scores higher with ipTM = 0.55 and pTM = 0.52, borderline results. Importantly, the topscoring complex of Spc105421-917-Kre28 is now different than the original model of Spc105490-917-Kre28. The top 5 models are not consistent and all contain multiple topological 'knots' indicative of erroneous predictions as mentioned above. Importantly, inclusion of Ipl11-30 models this fragment in the same location as the authors observe on a hybrid interface formed by Spc105 and Kre28 that is indeed similar in the top five scoring models. However, in the 5 topscoring models the rest of Spc105-Kre28 is variable, again giving very little confidence in the validity of these models. Importantly, in AlphaFold, even extending the Ipl1 fragment by one amino acid places Ipl11-31 in a different orientation rotated 180 degrees on the same region on Spc105421-917-Kre28. Taking the entire unstructured Ipl1 N-terminus (1-90) or full length Ipl1 again gives very different and inconsistent models. These results, together with the clear faults in the Spc105421-917-Kre28 model leave me with little to no confidence in these models, they should not serve as a basis for understanding the molecular details of the Spc105/Kre28 interaction with Sli15/Ipl1.

The authors do demonstrate the Ipl1 N-terminus may play a role in the interaction with Spc105/Kre28 in a pulldown experiments. However, I do not find the single Western showing moderately reduced Sli15 convincing. Where is Ipl1 in these pull downs (Fig 7C)? I would also like to see this on a Western. Also, could the authors pull down Spc105/Kre28 with Ipl1 N-terminal fragments, for example GST-Ipl11-30? The authors should further demonstrate these interactions using more quantitative biophysical methods such as ITC or biacore? Ultimately, it remains unclear what the function of the Ipl1 N-terminus is seeing also the minor effects, especially in the 3A mutant (Figs 7E-G).

Taken together, the authors demonstrate a novel Kre28 allele that results in reduced Spc105/Kre28 at kinetochores with a

concomitant reduction in error correction. They identify the Sli15-Ipl1 complex as a binding partner for Spc105/Kre28 and demonstrate this interaction is phosphoregulated by Ipl1. Importantly, the authors suggest that the reduction of Spc105/Kre28 for Kre28 Δ Zwift results in reduced Sli15-Ipl1 at kinetochores and hence, reduced error correction. I think this point can be strengthened by several experiments suggested above. I am less convinced by the experiments on the N-terminus of Ipl1 which appear to be largely driven by very unreliable alphafold models. I think, in general, the alphafold models have no place in this manuscript based off of the serious red flags mentioned above.

-
Minor points:

The authors claim the Zwift helix is among the most conserved segments of the protein. However, inspection of the sequences in Fig 1B shows no obvious sequence conservation with the colors that indicate biophysical properties of the underlying aa misleading. The authors should report sequence identity of the alignment to substantiate any claim that this sequence is conserved. Importantly, I was unable to reproduce the alignments shown in Fig 1B using MAFFT. The alignments within yeast are of course must more convincing (Fig 1, Sup 2)

The same holds true for the alignment in Fig 7A. The N-terminal tails shows a very small region that appears conserved, from 1-12. The rest is clearly not conserved. Importantly, two out of three mutants are derived from this very poorly conserved region? Yet again, the authors claim this segment is "well conserved among yeast aurora kinases".

In fig 4D the Spc105 pull downs show a band running at the height of Nuf2 which shouldn't be in the sample. Or is this instead HC, despite running lower? If this is the case, then what is the band in the Ndc80 /Mtw1 pulldowns that runs slightly higher? Nuf2 or HC? If that is Nuf2 like you would expect then where is the prominent band for HC in these samples? I am confused, the HC does not appear consistent between the samples.

In the TurboID experiments the authors compare biotinylation of Sli15-, Kre28- and Cin8-Flag in Spc105-TurboID expressing cells, showing no modification (no proximity) of Cin8. The authors claim: "despite similar input levels in extracts". The first thing I see is clearly significantly higher expression of Sli15-Flag compared to Cin8-Flag. These statements feel a bit disingenuous. I don't think it is a deal breaker for the experiment perhaps better to let words more accurately reflect the real situation.

The authors claim the truncated form of Sli15 purifies away from the Sli15/Ipl1 complex. That is not my interpretation of the gel in Fig 4 Supplement 1B. The gel filtration column clearly separates the larger FL Sli15 from the 'truncated' variant. However fractions B2-3, while containing mainly Sli15 Δ C still contain ample Ipl1. One could argue almost stoichiometric to Sli15 Δ C. See also comment below.

In the experiment to prove an important role for The Ipl1 N-terminus the authors actually show that now, in contrast to previous pulldowns Spc105/Kre28 can pull down the C-terminal truncation of Sli15 that should have no Ipl1 bound (Fig 7C). Could the authors comment on this?

In the discussion the authors suggest a model where the CPC may bind to both the centromere (via Sli15 N-terminal region) and the outer kinetochore (via Ipl1/Sli15 C-terminal segment). But is this true? Intra-kinetochore distances have been extensively mapped making it possible to estimate if the CPC/Sli15 could in fact bridge these distances. If the authors posit such a model, they should be able to at least assess its possible validity by making use of available distance restraints.

References

- Dabrowski-Tumanski P, Stasiak A (2023) AlphaFold Blindness to Topological Barriers Affects Its Ability to Correctly Predict Proteins' Topology. *Molecules* 28
- Hou Y, Xie T, He L, Tao L, Huang J (2023) Topological links in predicted protein complex structures reveal limitations of AlphaFold. *Commun Biol* 6: 1098
- Lensink MF, Brysbaert G, Raouraoua N, Bates PA, Giulini M, Honorato RV, van Noort C, Teixeira JMC, Bonvin A, Kong R et al (2023) Impact of AlphaFold on structure prediction of protein complexes: The CASP15-CAPRI experiment. *Proteins* 91: 1658-1683

Response to Reviewers: EMBOJ-2024-119215; Dudziak et al.**General remarks**

We thank all three reviewers for the positive evaluation of our study and the insightful comments on remaining open points.

Main criticism

The main criticism from all three reviewers was that the molecular nature of the binding site of Ipl1/Sli15 on Spc105/Kre28 has remained incompletely defined in the original version of the manuscript. The AlphaFold model predicting the binding of an N-terminal Ipl1 segment is of low confidence, the phenotypes of N-terminal Ipl1 mutants are consistent with a role in bi-orientation, but are not definitive and therefore the link to the original zwiint helix deletion has not been clear.

Major point for revision

We have discovered that the **Sli15 segment aa 215-255** is predicted by AlphaFold2 and 3 with much higher confidence on the helix bundle domain of Spc105/Kre28 (**new Figure 7, Figures EV4 and EV5**). The predicted binding mode of this Sli15 segment to the Spc105/Kre28 helix bundle characterized in our study is fully in line with the following points:

- 1)** Our crosslink-mass spec data of recombinant Ipl1-Sli15 bound to KMN presented in Figure 6.
- 2)** The previously reported importance of this Sli15 segment for bi-orientation in cells (Campbell and Desai, Nature, 2013) and our own mutational analysis presented in the **new Figures 7 and EV5**.
- 3)** an unpublished Spc105 mutant allele isolated in a screen for mutants specifically defective in tension-sensing/signaling (Spc105 E510K, Andrew Murray personal communication). Remarkably, this mutant was isolated in the same screen that has identified a null mutant in Sgo1 (Indejan and Murray, Science 2005).

We have constructed new yeast strains and generated new tools for biochemical experiments to evaluate the importance of this Sli15 segment and the binding interface. We have replaced the *SLI15* gene with *sli15* point mutants predicted to disrupt binding to the Spc105/Kre28 helical bundle, both in the context of the full-length Sli15 or in the Sli15 deltaN mutant previously characterized (Campbell and Desai, 2013). Our results indicate that Sli15-4E (R231E R232E L239E K242E) is a lethal mutation in haploid yeast cells, both in the context of the full-length protein and in the delta N mutant (**Figure 7E, Figure EV5**). In biochemical pull-down experiments the Sli15-4E mutant reduces binding to Spc105-Kre28. In a *sli15-aid* system, we find that the respective mutants are unable to rescue the depletion, with differential effects shown by specific mutations (**Figure EV5A-D**). Analyzing the localization of the mutant Sli15-4E in diploid yeast cells, we find that the 4E mutation causes a defect in kinetochore localization that is revealed in the context of Sli15 deltaN (**Figure 7F-G**).

In the revised manuscript we therefore present a **new Figure 7**, together with the **Expanded view figures EV4 and EV5** that summarize these exciting new findings. We have instead removed the analysis of the Ipl1 N-terminus from the study. We feel that the new Sli15 data tie together the study and provide convincing explanations for our original phenotypes.

Specific answers to reviewers' questions

Referee #1:

Dudziak et al. present extensive data documenting the presence of a new binding site for the Aurora B/Ipl1 kinase in the budding yeast kinetochore. The Aurora B kinase, hereon referred to as Ipl1, is the master regulator of the error correction pathway, which dissolves erroneous or syntelic chromosome attachments to the spindle. It has long been theorized that the error correction pathway is mediated by Ipl1 recruitment sites at the centromere and the centromere-proximal regions of the kinetochore. The new data strongly suggest that the Spc105-Kre28 complex located in the "outer" kinetochore harbors an additional Ipl1 binding site that makes significant contributions error correction in budding yeast. Thus, these findings will influence future models of the error correction mechanism and should generate strong interest.

We thank this reviewer for the positive evaluation of our study.

The authors are to be commended for presenting extensive and multi-pronged data to discover and characterize the interaction between the Spc105-Kre28 and Ipl1-Sli15 complexes. They offer conclusive evidence showing that: (a) the coiled-coil section of the Spc105-Kre28 complex is important for accurate chromosome segregation in a manner consistent with establishing proper Ipl1 activity in the kinetochore, (b) the Spc105-Kre28 can interact with the Ipl1-Sli15 complex in vitro, and this interaction is sensitive to Ipl1 autophosphorylation, (c) artificial recruitment of Ipl1 in a similar site as Spc105-Kre28 restores cell viability in Kre28 mutants. Overall, the experiments are well-designed and executed but, the picture that emerges contains some significant gaps discussed below. Nevertheless, this work achieves a significant advance in our understanding of the error correction mechanism; it will guide future efforts to fully defining the mechanism of error correction. Therefore, I support its publication after the authors have addressed the following issues.

Main comments:

1. What is the exact nature of the Ipl1/Sli15 binding site in Spc105-Kre28? The evidence for the new Ipl1-Sli15 binding site in the Spc105-Kre28 complex is not definitive. While the authors perform extensive and careful pull-down experiments with recombinant proteins, I am unable to rationalize the following incongruency. The conserved "Zwint-1" helix in Kre28 is clearly important for accurate chromosome segregation, however, the AlphaFold predictions (Fig. 1) show that it is located distal to the interaction site for Ipl1 predicted in Fig. 2. Interestingly, deletion of the Zwint-1 helix does not appear to have any effect in the in vitro pull-down assay shown in Figure 4c. Furthermore, the chemical crosslinking experiment also did not yield any hits with this section of Zwint-1. The authors should offer the best

possible explanation for why this might be so. For example, does the deletion of this helix simply disrupt the Ipl1-binding interface? Is the reduced amount of Spc105 in this mutant result in reduced Ipl1 recruitment? I suggest that the authors fully discuss the implications of their findings to help the reader in achieving a good understanding of the work.

As stated in our overall remarks, we discovered that Sli15 segment 215-255 is predicted with much higher confidence by AlphaFold, and we have analyzed mutations based on this new prediction. This prediction can also explain why the original Zwint helix deletion does not have a stronger impact: Only Kre28 residues 90-112 seem to be involved in binding the Sli15 peptide, the rest of the helix is located on the other side of the binding interface. This is shown in the new Figure EV4 C. This is also fully consistent with our mutational analysis in Figure EV1.

2. The synthetic interactions of the Ipl1 N-terminal mutants are intriguing and consistent with the authors' model. However, it remains possible that other Ipl1 interactions mediated by the residues are responsible for the phenotype. The authors should test whether artificial recruitment of Ipl1 used earlier can rescue viability of the *ipl1-3A ctf19-RWDdelete* mutant.

We have removed the Ipl1 N-terminus data from the revised manuscript in favor of the much stronger Sli15 data.

3. In the same vein, it is important to perform one additional experiment because AF2 predicts the Ipl1-Spc105 interaction with very low confidence. The authors should disrupt the binding interface in Spc105 by mutating residues predicted by the AF2 models to interact with the Ipl1 N-terminus. Genetic interactions of this mutant with Sgo1-AID or *ctf10-RWDdelete* mutants will significantly strengthen the authors' case.

We have constructed mutations in Spc105 residues predicted to be involved in binding the Sli15 segment. We find benomyl-hypersensitivity and a strong growth defect, in accordance with the corresponding *sli15* mutations (Figure EV5 E,F). Our mutational analysis included Spc105 E510K, a mutant previously isolated a screen designed to identify lack of tension sensing/signalling (Andrew Murray, personal communication).

4. A predicted structure of Ipl1 and Spc105-Kre28 that highlights the interaction surface on Spc105-Kre28 as well as the location of the Zwint-1 helix in Kre28 would be very useful to help the readers understand the nature of the putative interaction site. Surface color depicting a conservation score would also be quite useful to understand whether the yeast results can be generalized easily to other eukaryotes.

We provide the first point in the new Figure 7A, B and Figure EV4C. We are hesitant to comment on general conservation. The organization of the corresponding human Knl1-Zwint helix bundle is predicted to be similar (Polley et al., NSMB 2024), but INCENP predictions on it are not quite as convincing as with the yeast proteins (our unpublished observations). Without functional data we feel it is therefore premature to comment on general conservation.

Minor comments:

1. Is it possible that the two appendages in Fig. 4B correspond to the MIND and Spc105-Kre28 coiled-coil domains respectively?

It's possible, but hard to state definitively. We have added a sentence speculating about this in the revised manuscript.

2. Indicate the coiled coil domains of Spc105 and the Zwint-1 helix in Kre28 in Figure 5.

We have indicated this in the revised Figure 6.

3. Label the bands corresponding to Ipl1 and Sli15 in Figure 7.

We have done this in the revised Figure 7D.

4. The PAE plots in the supplementary Figure 7 are too small to study especially for the small section of Ipl1. The PAE plot for the best model should be presented by itself and made legible.

We provide a new single PAE plot for the highest ranking model of Spc105-Kre28-Sli15 in the new Figure EV4 A.

5. It would be useful to use the colony color assay to evaluate the contributions of the Zwint-1 helix to accurate chromosome segregation under normal conditions. This is optional because it adds context that is not necessary to support the main advance of this work.

Given the much stronger phenotypes provided by the *sli15* mutants in this interface in general, we have postponed this experiment for the time being.

Referee #2:

The manuscript "The Spc105Knl1/Kre28Zwint complex promotes mitotic error correction by outer kinetochore recruitment of the Ipl1Aurora B kinase" by Dudziak et al explores an important topic for the kinetochore and mitosis fields. Determining the outer kinetochore receptor of Ipl1 is a significant topic and would provide needed insight into mechanisms of kinetochore-microtubule error-correction in particular. In their manuscript, Dudziak and colleagues begin by characterizing a kre28 mutant in a conserved alpha helical region proximal to the Spc105-binding alpha helical domain. A deletion of this "zwint helix" renders cells sensitive to cold temperatures and the microtubule poison benomyl and has phenotypes consistent with defective mitotic error correction, including synthetic viability defects with *ipl1-1* and chromosome segregation defects. They also show that this *kre28Δzwint* mutant has a proficient SAC but results in reduced Kre28 and Spc105 recruitment to kinetochores. They further show that Ipl1/Sli15 can directly bind to Spc105/Kre28 in vitro to localize the kinase to the KMN network. Crosslinking mass spectrometry confirmed an interaction between Ipl1, Sli15, and Spc105/Kre28 in vitro. Significantly, they showed that a Mtw1-Sli15 fusion was able to rescue the phenotypes of

kre28 Δ zwint, in support of the notion that Spc105/Kre28 recruit Ipl1 to the kinetochore. They further propose that the Ipl1 N-terminus comprises the Spc105/Kre28 binding module and present ipl1 mutants with biochemical and cell phenotypes that are consistent with this idea. Overall, the authors propose a reasonable model that is supported by their data. They are very careful in their writing and do not overclaim what their data show. The Mtw1-Sli15 fusion rescue experiments are superb and are the highlight of the paper, and by contrast many of the biochemical assays are weaker and less clear. However, we recognize the sheer number of biochemical assays shown, and the work this represents, and note that the phenotypes while weak are at least mostly consistent with their model. The data shown in in Figure 7 is weak, in particular the AlphaFold 2 prediction shown in more detail in the supplement, but their mutational analysis is consistent with the conclusions they draw.

We thank this reviewer for the positive evaluation of our study. We acknowledge that the biochemistry has been challenging. These are complicated multi-protein complexes and some subunits, such as Spc105, are particularly difficult to work with. An important point for the future is develop more quantitative binding assays, which should now be possible with some of the key interfaces identified.

Major points:

1. The structural prediction in Figure 7B is not of high confidence based on the pLDDT metrics and aligned error provided in the supplement. The multiple predicted binding modes shown also do not provide confidence that this structure prediction is correct. I suggest the authors try this prediction again with a larger region of Ipl1 to exclude the possibility that another region contributes to Spc105/Kre28 association. In the absence of this, perhaps a prediction carried out using AlphaFold 3 might also result in a higher confidence model. Alternatively, demonstrating that mutations on the predicted Spc105/Kre28 side of the interface produce a similar phenotype to the ipl1 mutants would strengthen the authors' claim.

Please see our general remarks. The Sli15 peptide provides a much higher confidence prediction. We now also analyze mutations in Spc105, as suggested by this reviewer (new Figure EV5).

2. The mutational analysis in Figure 7E-G does not strictly support the notion that the Ipl1 N-terminal region binds to the outer kinetochore. It is possible that simply combining multiple slightly deleterious mutations in different pathways results in more severe phenotypes. A rescue of an IPL1 mutant via tethering to MTW1 as in Figure 6C would provide better support to their conclusions.

We agree. We have removed the Ipl1 data in favor of the stronger Sli15 analysis.

Minor Points

1. The order of presentation of the data leads a reader to think early on that kre28 Δ zwint may specifically ablate Ipl1 binding at the kinetochore. This leads to confusion in Figure 4 when it becomes apparent at Kre28 Δ zwint can bind Ipl1 similar to WT in vitro. While the authors are careful in stating the loss of Ipl1 in kre28 Δ zwint is due to reduced Mtw1-

Spc105/Kre28 association, I think a reordering of the data would better convey what they are trying to say. I propose a reordering of the data to place the figures as written in a new order (Figure 1, 2, 6, 4, 5, 7). Moving the Figure 6 data closer to Figure 2 will help better convey the kre28 Δ wint phenotypes are due to loss of Ipl1, and that this loss is likely due to reduced association of Spc105/Kre28 within KMN. The authors should clarify that they aim to identify a more specific mutant that effectively delocalizes Ipl1 from Spc105/Kre28 to test their prediction more robustly. Finding such a mutant - one that maintains KMN assembly while excluding Ipl1 - should be noted as a priority, even if not essential for the current study.

We have followed this suggestion and moved figure 6 (rescue experiment with Sli15 fusions) closer to figures 2 and 3. This new order now works better, because now the new figure 6 (crosslinking) is connected better to new Figure 7 (Sli15 binding site).

2. The crosslinking mass spectrometry data in Figure 5 do not show any interaction between the Ipl1 N-terminal region and Spc105/Kre28. In line 453 the authors state the crosslinking data supports the notion that the Ipl1 N-terminal region contacts Spc105/Kre28. They should remove this language and comment on why they think such crosslinks do not exist if they propose this region to mediate Ipl1-KMN interaction. In addition, this crosslinking figure seems disconnected from the rest of the manuscript and could also be reordered to be in or near Figure 1 in support of the notion that Spc105/Kre28 recruit Ipl1.

Agreed. We have removed the Ipl1-N statement. As stated before, the crosslinking data is now much better connected to the Sli15 binding site (new Figures 6 and 7). We highlight Sli15 K217 in the Sli15 alignment in Figure 7C.

3. While very exhaustive, some biochemical analyses show weaker than expected phenotypes especially given the robust cellular response to mutational analysis. In particular, Figures 3C-D appear relatively weak compared to the strong loss of Spc105/Kre28 at the kinetochore observed in cells. While the quantification in 3D helps, the gel itself is not very compelling (for example, the Mtw1 band shows little noticeable difference). Minimally, the authors should comment on why more pronounced differences were not observed. Additionally, there is disagreement between Figure 4C and the accompanying supplement with regards to Ipl1 binding. We propose the authors move 4C to supplement and instead rely on Figure 4D and 4F to conclude Spc105/Kre28 recruit Ipl1 to KMN in vitro.

We agree and have removed Figure 3C and D.

4. If Kre28 Δ wint disrupts binding to Mtw1, but still binds Ipl1, you might expect in vivo biotinylation with Spc105-TurboID to show labeling of Sli15 but not an Mtw1-FLAG construct. Completing this experiment would lend confidence to weaker in vitro results.

Given the Sli15 data, we feel this is not necessary at this point.

5. Can a direct Ipl1 in vivo localization assay be performed for any of the mutants tested? Perhaps a Co-IP or fluorescence microscopy.

We have done this now for the Sli15 and the Sli15-4E mutant with mNeonGreen fusions (new Figure 7 F,G). A clear difference can be observed in the context of the Sli15-deltaN constructs.

6. In Figure 2E is it possible to tell whether mis-segregation had any preference for the mother or daughter cell? Providing this information would also provide confidence that kre28 Δ zwint phenotypes are consistent with an ipl1 mutant phenotype.

We didn't notice a bias, but in this type of nocodazole wash-out/spindle reformation assay after a long arrest, we would not necessarily expect this, even if the defect is ipl1-dependent.

7. In Figure 6C Sli15 point mutants fusions should be on the same plate as the Sli15-WT fusions.

Following this suggestion, we have modified the new Figure 4C to include this on the same plate.

8. Very minor, can the authors mention why their sgo1-AID strain is using pGAL-TIR?

This experiment has been removed from the manuscript. (for interest: this yeast strain contained an older system that used conditional TIR expression for tighter control and more complete depletion. We have now switched to constitutive TIR expression under strong promoters).

9. Can the authors label the Zwint helix on Figure 5 and Figure 7? Also, can the authors label residue numbers on Ipl1 in Figure 5?

We provide this in the new Figure 6 and Figure EV4C.

10. Line 108: typo "INCEP" should be "INCENP"

Thank you, corrected!

Referee #3:

Dudziak et al describe a novel allele of yeast Kre28 that highlights a role for Spc105/Kre28 in error correction by facilitating Ipl1 recruitment to kinetochores. The manuscript is very well written, suitable for a broad audience. The work is of high importance since, despite a wealth of studies, how exactly Aurora B regulates error correction remains incompletely understood. Recent studies have highlighted multiple pools of the CPC besides the classical inner centromere and the presence of an (outer)-kinetochore pool has been suggested. However, how the CPC interacts with outer kinetochore has remained elusive. Identifying

Spc105/Kre28 as a possible receptor opens a new avenue of research to investigate how the CPC regulates kt-mt attachments, both in yeast as in mammals. However, in its current form the manuscript also contains some serious shortcomings undermining the molecular understanding of the identified Kre28 Δ Zwint allele.

We thank the reviewer for the positive evaluation and the recognition of the importance. We think the molecular understanding is now significantly improved with the analysis of the sli15 mutants.

The authors model the Spc105-Kre28 complex using alphafold2. Using S.c. Spc105490-917 and FL Kre28 I was able to reproduce the structure they present in Fig 1A. Importantly, this structure scores poorly with pTM and ipTM scores of 0.44. Per alphafold: "A pTM score above 0.5 means the overall predicted fold for the complex might be similar to the true structure." "ipTM measures the accuracy of the predicted relative positions of the subunits within the complex. Values higher than 0.8 represent confident high-quality predictions, while values below 0.6 suggest likely a failed prediction." Moreover, closer inspection of the model reveals multiple topological intertwinements that effectively do not appear in experimentally determined structures. This is a known problem for AlphaFold predictions of complexes associated with poor model quality, see for example refs (Dabrowski-Tumanski & Stasiak, 2023; Hou et al, 2023; Lensink et al, 2023). Taken together, the presented structural model is very likely incorrect and inspires little confidence as a molecular basis for interpreting interactions.

These are all fair points, and we thank the reviewer for pointing out the limitations of the structure predictions. The new prediction of the Spc105-Kre28-Sli15 assembly, presented in the new Figure 7 and Figure EV4, scores significantly better with pTM and ipTM scores over 0.85 (high confidence) and very reasonable pLDDT scores for the central part of the Sli15 segment.

The authors proceed to focus on a helical fragment in Kre28 they name the Zwint helix (residues 77-113). Importantly, the Zwint helix highlights the above mentioned problems with topological intertwining as residues (114-126), C-terminal of the Zwint helix thread through a looped region formed by Spc105 residues 490-640.

Regardless, deletion of the Zwint helix in combination with the RWD domain in Kre28 impairs viability and Kre28 Δ Zwift confers benomyl hypersensitivity and cold sensitivity to cells, indicating a possible role in error correction. Importantly, the authors demonstrate an interaction between the KMN network and the error correction kinase Ipl1 in complex with Sli15.

The authors demonstrate that the recombinant Ipl1/Sli15 complex is (auto)phosphorylated when purified from E. coli. They then demonstrate this autophosphorylated complex binds to the KMN network, specifically to the Spc105/Kre28 subcomplex (Fig 4C,D). Thus, clearly autophosphorylated Sli15-Ipl1 can bind to the KMN network. Now, the recombinant complexes are incubated with ATP or lambda phosphatase and after 30 min at 30 degr. 100 ul (per methods section 'Pull down assays') Ipl1/Sli15 mix is added to 20 ul preloaded (with

KMN network) beads. These beads are now incubated for another 6 hours at 4 degr prior to wash and elution. It is unclear how the authors now conclude that autophosphorylation of Ipl1/Sli15 is the sole reason for diminished binding in the ATP samples. During incubations it is highly likely that the KMN network is phosphorylated in the ATP samples. I don't think the authors can rule out that KMN phosphorylation contributes to the diminished binding of Ipl1/Sli15. To prove it is specifically autophosphorylation of Ipl1/Sli15 would require this complex to be further purified after ATP/Lambda phosphatase incubation such that the ATP/Lambda phosphatase only 'sees' Ipl1/Sli15. It is also unclear what the phosphorylation status of the KMN prep from insect cells is? Importantly, phosphorylation of KMN and autophosphorylation of Ipl1/Sli15 need not be mutually exclusive as a mode of regulating an Ipl1/Sli15 pool at the kinetochore.

These are all valid points. We do not know the precise phosphorylation state of the Spc105/Kre28 complex when purified from insect cells and we can't exclude effects of changing its phosphorylation state in the pull-down experiment. We have added a sentence to this effect (line 413f) . Investigation of regulatory mechanisms will require us to develop more quantitative binding assays, which is an important goal for future experiments.

The authors reason that Spc105/Kre28 is important for Sli15/Ipl1 recruitment to kinetochores. Importantly, Kre28 Δ Zwift results in reduced levels of Spc105/Kre28 and hence Sli15/Ipl1 at kinetochores. However, constitutive recruitment of Sli15/Ipl1 to kinetochores via fusion to Mtw1 only partially rescues missegregation rates in cells expressing the Kre28 Δ Zwift allele. The authors should really emphasize the partial nature of the rescue here. Clearly, the fusion is not nearly enough with close to 40% of cells still missegregating chromosome XII. Importantly, the authors demonstrate Sli15/Ipl1 recruitment to kinetochores is likely phosphoregulated, thus constitutive recruitment may not be expected to fully rescue the Kre28 Δ Zwift allele as it does not permit a dynamically regulated pool of Sli15/Ipl1. This needs to be discussed more clearly. Furthermore, how do Sli15/Ipl1 levels change at kinetochores between WT, Kre28 Δ Zwift and Mtw1-Sli15? Importantly, can the authors rule out that placing 'extra' Ipl1 activity results in a specific rescue by increasing MT turnover at kinetochores? To drive home the point that loss of Spc105/Kre28 in Kre28 Δ Zwift cells results in loss of a kinetochore pool of Sli15/Ipl1 the authors should demonstrate a rescue by reconstituting kinetochore levels of Spc105/Kre28, for example by tethering Kre28 or Kre28 Δ Zwift to Mtw1.

We have added additional sentences discussing the partial effect of the rescue experiments (lines 398 ff). The reviewer is correct that from a biological perspective, it is difficult to envision that a full rescue with such a constitutive fusion should even be possible. Besides, there is the technical limitation that the fusion is expressed over a wild-type Mtw1, with which it competes for kinetochore localization.

The authors further focus on a potential role of the Ipl1 Nterminus in the interaction with Spc105-Kre28, based in part on previous data that suggest a region around residue 45-55 in binding to Ndc80. Again, the authors use AlphaFold modeling to study the how Ipl1 1-30 binds to Spc105-Kre28 but now also in the context of the kKMN network. Importantly, now a larger fragment of Spc105 is chosen, residues 421-917. This complex scores higher with

ipTM = 0.55 and pTM = 0.52, borderline results. Importantly, the topscoring complex of Spc105421-917-Kre28 is now different than the original model of Spc105490-917-Kre28. The top 5 models are not consistent and all contain multiple topological 'knots' indicative of erroneous predictions as mentioned above. Importantly, inclusion of Ipl11-30 models this fragment in the same location as the authors observe on a hybrid interface formed by Spc105 and Kre28 that is indeed similar in the top five scoring models. However, in the 5 topscoring models the rest of Spc105-Kre28 is variable, again giving very little confidence in the validity of these models. Importantly, in AlphaFold, even extending the Ipl1 fragment by one amino acid places Ipl11-31 in a different orientation rotated 180 degrees on the same region on Spc105421-917-Kre28. Taking the entire unstructured Ipl1 N-terminus (1-90) or full length Ipl1 again gives very different and inconsistent models. These results, together with the clear faults in the Spc105421-917-Kre28 model leave me with little to no confidence in these models, they should not serve as a basis for understanding the molecular details of the Spc105/Kre28 interaction with Sli15/Ipl1.

We fully agree with the criticism of the original Ipl1-N prediction. In our view the new prediction of the Sli15 peptide both provides a more convincing model and yields more penetrant phenotypes that are consistent with an important role in error correction.

The authors do demonstrate the Ipl1 N-terminus may play a role in the interaction with Spc105/Kre28 in a pull-down experiments. However, I do not find the single Western showing moderately reduced Sli15 convincing. Where is Ipl1 in these pull downs (Fig 7C)? I would also like to see this on a Western. Also, could the authors pull down Spc105/Kre28 with Ipl1 N-terminal fragments, for example GST-Ipl11-30? The authors should further demonstrate this interactions using more quantitative biophysical methods such as ITC or biacore? Ultimately, it remains unclear what the function of the Ipl1 N-terminus is seeing also the minor effects, especially in the 3A mutant (Figs 7E-G).

We provide new pull-down assays with the Sli15-4E mutant, which shows a partial reduction, but also not a complete elimination of binding. We envision that the Sli15 segment 215-255, while crucial for function, may not be the only Sli15 segment capable of binding KMN. In line with this, we note that Sli15-4E also does not seem to completely eliminate kinetochore localization in the deltaN mutant, consistent with the existence of additional binding interfaces. Future work will have to clarify the binding mode systematically. We fully agree that for this, more quantitative binding assays will need to be developed.

Taken together, the authors demonstrate a novel Kre28 allele that results in reduced Spc105/Kre28 at kinetochores with a concomitant reduction in error correction. They identify the Sli15-Ipl1 complex as a binding partner for Spc105/Kre28 and demonstrate this interaction is phosphoregulated by Ipl1. Importantly, the authors suggest that the reduction of Spc105/Kre28 for Kre28 Δ Zwift results in reduced Sli15-Ipl1 at kinetochores and hence, reduced error correction. I think this point can be strengthened by several experiments suggested above. I am less convinced by the experiments on the N-terminus of Ipl1 which appear to be largely driven by very unreliable alphafold models. I think, in general, the

alphafold models have no place in this manuscript based off of the serious red flags mentioned above.

We respectfully disagree with the last point and think that especially the high confidence AlphaFold model provided with the Sli15 segment is important to inspire and design specific mutations that interrogate the importance of this interface. We show that model-guided point mutations in Sli15 impair viability and reduce the localization of Sli15 to kinetochores in cells (revealed in the deltaN mutant). Corresponding point mutations in Spc105 display complementary phenotypes. The model also delivers an explanation for why the original zwiint helix deletion appears to be a hypomorphic mutation: only the c-terminal part of the helix contributes to the Sli15 binding site, the rest is contributed by Spc105. Following the suggestion, we removed other more extensive alpha fold predictions (for example the spliced KMN model), which at this point do not contribute direct insights.

-

Minor points:

The authors claim the Zwiint helix is among the most conserved segments of the protein. However, inspection of the sequences in Fig 1B shows no obvious sequence conservation with the colors that indicate biophysical properties of the underlying aa misleading. The authors should report sequence identity of the alignment to substantiate any claim that this sequence is conserved. Importantly, I was unable to reproduce the alignments shown in Fig 1B using MAFFT. The alignments within yeast are of course must more convincing (Fig 1, Sup 2)

The alignment in Figure 1B was generated using homology searches and secondary structure predictions as described in (Schleiffer et al., 2012). Predictions of the human Knl1/Zwiint structure indicate a similar helix donated by Zwiint-N to a 4-helix bundle with Knl1 (Polley et al., NSMB 2024). See also Tromer et al., PNAS 2019 for the conserved organization of Zwiint proteins.

The same holds true for the alignment in Fig 7A. The N-terminal tails shows a very small region that appears conserved, from 1-12. The rest is clearly not conserved. Importantly, two out of three mutants are derived from this very poorly conserved region? Yet again, the authors claim this segment is "well conserved among yeast aurora kinases".

The Ipl1 alignment has been replaced by the Sli15 alignment, we have adjusted the wording. The sli15-4E point mutant now indeed affects conserved residues.

In fig 4D the Spc105 pull downs show a band running at the height of Nuf2 which shouldn't be in the sample. Or is this instead HC, despite running lower? If this is the case, then what is the band in the Ndc80 /Mtw1 pulldowns that runs slightly higher? Nuf2 or HC? If that is Nuf2 like you would expect than where is the prominent band for HC in these samples? I am confused, the HC does not appear consistent between the samples.

Thanks for noticing this and bringing it to our attention These are peptide elutions, so HC should not be present in pull-down elutions and indeed one doesn't see LC. The same phenomenon observed by the reviewer can also be seen in 4E, where the additional band is clearly distinct from HC in the input. Judged by the relative intensities of full-length Spc105 band and the Kre28 band, it may be a proteolysis product of full-length Spc105. The N-terminus is disordered, a stable fragment lacking the N-terminus could be some 50-55 kDa in size, which would roughly fit to the observed band. We have added an asterisk to the Figure description, making aware of this point. We note that this would not affect any conclusions.

In the TurboID experiments the authors compare biotinylation of Sli15-, Kre28- and Cin8-Flag in Spc105-TurboID expressing cells, showing no modification (no proximity) of Cin8. The authors claim: "despite similar input levels in extracts". The first thing I see is clearly significantly higher expression of Sli15-Flag compared to Cin8-Flag. These statements feel a bit disingenous. I don't think it is a deal breaker for the experiment perhaps better to let words more accurately reflect the real situation.

Fair point. We have changed the wording to acknowledge the lower expression level of Cin8-Flag (line 305f).

The authors claim the truncated form of Sli15 purifies away from the Sli15/Ipl1 complex. That is not my interpretation of the gel in Fig 4 Supplement 1B. The gelfiltration column clearly separates the larger FL Sli15 from the 'truncated' variant. However fractions B2-3, while containing mainly Sli15 Δ C still contain ample Ipl1. One could argue almost stoichiometric to Sli15 Δ C. See also comment below.

We have removed this supplementary figure from the revised manuscript. Indeed, we could not consistently purify away the shorter Sli15 fragment from Ipl1, so different preparations might contain different amounts of this contamination. This will have to be improved in future purification schemes.

In the experiment to prove an important role for The Ipl1 N-terminus the authors actually show that now, in contrast to previous pulldowns Spc105/Kre28 can pull down the C-terminal truncation of Sli15 that should have no Ipl1 bound (Fig 7C). Could the authors comment on this?

This has been removed from the revised manuscript.

In the discussion the authors suggest a model where the CPC may bind to both the centromere (via Sli15 N-terminal region) and the outer kinetochore (via Ipl1/Sli15 C-terminal segment). But is this true? Intra-kinetochore distances have been extensively mapped making it possible to estimate if the CPC/Sli15 could in fact bridge these distances. If the authors posit such a model, they should be able to at least asses its possible validity by making use of available distance restraints.

Only the N-terminus of Sli15 (binding Bir1/Nbl1) and the single alpha helix domain (SAH) are predicted to form alpha helices. The vast majority of this almost 700 residue polypeptide is predicted to lack secondary structure. The maximum distances between inner and outer kinetochores have been estimated to around 80-90 nm in yeast (Joglekar et al. 2009). A largely disordered Sli15 coil could easily span more than 100 nm in solution. We therefore envision multiple connection points of Sli15 to inner and outer kinetochores.

References

- Dabrowski-Tumanski P, Stasiak A (2023) AlphaFold Blindness to Topological Barriers Affects Its Ability to Correctly Predict Proteins' Topology. *Molecules* 28
- Hou Y, Xie T, He L, Tao L, Huang J (2023) Topological links in predicted protein complex structures reveal limitations of AlphaFold. *Commun Biol* 6: 1098
- Lensink MF, Brysbaert G, Raouraoua N, Bates PA, Giulini M, Honorato RV, van Noort C, Teixeira JMC, Bonvin A, Kong R et al (2023) Impact of AlphaFold on structure prediction of protein complexes: The CASP15-CAPRI experiment. *Proteins* 91: 1658-1683

Prof. Stefan Westermann
University of Duisburg-Essen
Department of Molecular Genetics, Faculty of Biology
Centre of Medical Biotechnology
Essen 45117
Germany

27th Mar 2025

Re: EMBOJ-2024-119215R
Spc105/Kre28 promotes mitotic error correction by outer kinetochore recruitment of Ipl1/Sli15

Dear Stefan,

Thank you again for submitting your revised manuscript for our consideration. Two of the original referees have now assessed it once more, and I am happy to say that both were fully satisfied with the experimental and presentational revisions, and have no further concerns at this stage. After incorporation of the following remaining editorial issues, we should therefore be able to proceed with formal acceptance of the study:

- Please reduce the number of keywords on the abstract page to 5, ideally choosing general conceptual terms over particular gene names.
- Please carefully check the bibliography for completeness (journal, year, volume, pagination...) of all references. Also, please adjust the format for preprint citation as specified in our author guidelines. The citation in the text should be: "(PREPRINT: name1 et al, year)"; in the reference list: "Author name1, Author name2, ... (year) article title. bioRxiv doi: XXX"
- Please add at least one reference to Figure 4C, which seems to be currently missing (on page 8?).
- In the Source Data Checklist, please comment in the blue free-text boxes at the bottom on the meaning of the "XX" labels (esp. in the case of Figs/ 7E-G, for which source data has been provided), reasons for not including some of the requested source data (e.g. for Fig 3C-D) - please clarify.
- Finally, please provide suggestions for a short 'blurb' text prefacing and summing up the study in two sentences (max. 250 characters), followed by 3-5 one-sentence 'bullet points' with brief factual statements about key results of the paper; they will form the basis of an editor-written 'Synopsis' accompanying the online version of the article (see new articles on our journal website for some recent examples). Please also provide a simple synopsis image, which can be used as a "visual title" for the synopsis section of your paper (maybe a simplified/compacted version of Figure 8?). The image should be in PNG or JPG format with the modest dimensions of 550 x 300-600 pixels (width x height).

I am returning the manuscript to you for a final round of revision, solely to allow you to make these modifications and upload the revised files. Once we will have received them, we should be ready to swiftly proceed with formal acceptance and production of the manuscript.

With kind regards,

Hartmut

- 2) Each figure legend must specify
 - size of the scale bars that are mandatory for all micrograph panels
 - the statistical test used to generate error bars and P-values
 - the type error bars (e.g., S.E.M., S.D.)
 - the number (n) and nature (biological or technical replicate) of independent experiments underlying each data point
 - Figures may not include error bars for experiments with $n < 3$; scatter plots showing individual data points should be used instead.
- 3) Revised manuscript text (including main tables, and figure legends for main and EV figures) has to be submitted as editable text file (e.g., .docx format). We encourage highlighting of changes (e.g., via text color) for the referees' reference.
- 4) Each main and each Expanded View (EV) figure should be uploaded as individual production-quality files (preferably in .eps, .tif, .jpg formats). For suggestions on figure preparation/layout, please refer to our Figure Preparation Guidelines: <http://bit.ly/EMBOPressFigurePreparationGuideline>
- 5) Point-by-point response letters should include the original referee comments in full together with your detailed responses to them (and to specific editor requests if applicable), and also be uploaded as editable (e.g., .docx) text files.
- 6) Please complete our Author Checklist, and make sure that information entered into the checklist is also reflected in the manuscript; the checklist will be available to readers as part of the Review Process File. A download link is found at the top of our Guide to Authors: embopress.org/page/journal/14602075/authorguide
- 7) All authors listed as (co-)corresponding need to deposit, in their respective author profiles in our submission system, a unique ORCID identifier linked to their name. Please see our Guide to Authors for detailed instructions.
- 8) Please note that supplementary information at EMBO Press has been superseded by the 'Expanded View' for inclusion of additional figures, tables, movies or datasets; with up to five EV Figures being typeset and directly accessible in the HTML version of the article. For details and guidance, please refer to: embopress.org/page/journal/14602075/authorguide#expandedview
- 9) To facilitate reproducibility and cross-laboratory adoption of methodologies, please structure the Materials & Methods section as outlined in our guide to authors, including a completed Reagents and Tools Table that can be downloaded from our author guidelines as well (<https://www.embopress.org/page/journal/14602075/authorguide#structuredmethods>).
- 10) Digital image enhancement is acceptable practice, as long as it accurately represents the original data and conforms to community standards. If a figure has been subjected to significant electronic manipulation, this must be clearly noted in the figure legend and/or the 'Materials and Methods' section. The editors reserve the right to request original versions of figures and the original images that were used to assemble the figure. Finally, we generally encourage uploading of numerical as well as gel/blot image source data; for details see: embopress.org/page/journal/14602075/authorguide#sourcedata

At EMBO Press, we ask authors to provide source data for the main manuscript figures. Our source data coordinator will contact you to discuss which figure panels we would need source data for and will also provide you with helpful tips on how to upload and organize the files.

In the interest of ensuring the conceptual advance provided by the work, we recommend submitting a revision within 3 months (25th Jun 2025). Please discuss the revision progress ahead of this time with the editor if you require more time to complete the revisions. Use the link below to submit your revision:

Link Not Available

Referee #1:

The revised version presented by authors is significantly improved. Using a combination of yeast genetics, biochemistry, and AF3 structure predictions, the authors have compiled very convincing evidence for a new binding site for the Aurora-B kinase in the outer kinetochore in yeast. This finding brings into focus the mechanistic basis of error correction in budding yeast and,

likely, other eukaryotes.

I have no further recommendations!

Referee #2:

The revised manuscript "The Spc105/Kre28 complex promotes mitotic error correction by outer kinetochore recruitment of Ipl1/Sli15" by Dudziak and colleagues presents a much stronger collection of evidence for identifying the outer kinetochore localization module for Ipl1/Sli15. In the original submission, problems existed with weak biochemistry, and an especially poor AlphaFold prediction of Ipl1:Spc105/Kre28 interaction, and unclear mutational analysis of Ipl1 mutants. The authors have since removed the questionable AlphaFold prediction and ambiguous Ipl1 data in favor of a completely new figure showing Ipl1/Sli15 interaction with KMN is mediated instead by Sli15 directly interacting with Spc105/Kre28. This represents new analysis in response to reviewers' critiques that is much stronger than the original assertion centered on Ipl1:Kre28/Spc105 binding. Given the strength of the new analysis and the removal of problematic data, it is my feeling that the manuscript adequately addresses the reviewer criticisms.

General comments:

1. We still feel the rescue of kre28delzwint by tethering of Sli15 to Mtw1 is a fantastic experiment, and strongly supports the authors model of Sli15/Ipl1 kinetochore localization.
2. The new Sli15 AlphaFold prediction (Fig. 7A) is of much higher quality than the original Ipl1 prediction. Additionally, the mutational analysis of Sli15 mutants is much more consistent than the Ipl1 mutants, especially in pull down experiments. We note that some inputs in Fig. 7D appear "messy" and could show sharper bands, however, we do not feel this detracts from the main message that Sli154E reduces Ipl1/Sli15:Spc105 association. Additionally, the cellular phenotypes of Sli15 mutants are very strong, and consistent with loss of Ipl1 kinetochore function. If we could request one more experiment from the authors, it would be an attempted rescue of the new Sli15 mutants by tethering them to Mtw1 as in Fig. 4, either using the tethering system described or other analogous system. However, we do not feel this is necessary.
3. The authors rearrangement of the text helps the manuscript flow better and connects the related data in a more understandable manner. Thank you for taking the suggestion.